# Evaluating Snow Depth Retrievals from Sentinel-1 Volume Scattering over NASA SnowEx Sites

Zachary Hoppinen[1,2], Ross T. Palomaki[3], George Brencher[4], Devon Dunmire[5,6], Eric Gagliano[4], Adrian Marziliano[7], Jack Tarricone[8,9], and Hans-Peter Marshall[1]

[1]Boise State University, Department of Geosciences, 1295 University Drive, Boise, ID, USA
[2]Cold Regions Research and Engineering Laboratory, Engineer Research and Development Center, United States Army, Hanover, NH 03755, USA
[3]Institute of Arctic and Alpine Research, University of Colorado, 4001 Discovery Dr, Boulder, CO 80303
[4]Civil and Environmental Engineering Department, University of Washington, Seattle, WA 98195 USA
[5]Department of Earth and Environmental Sciences, KU Leuven, Heverlee, Belgium
[6]Department of Atmospheric and Oceanic Sciences, CU Boulder, 4001 Discovery Dr, Boulder, CO 80303
[7]Department of Civil, Construction, and Environmental Engineering, University of New Mexico, Albuquerque, NM 87131, USA
[8]Hydrological Sciences Laboratory, NASA Goddard Space Flight Center, Greenbelt, MD 20770, USA
[9]NASA Postdoctoral Program, NASA Goddard Space Flight Center, Greenbelt, MD 20770, USA

**Correspondence:** Zachary Hoppinen (Zachary.Hoppinen@u.boisestate.edu)

**Abstract.**

Snow depth retrievals from spaceborne C-band synthetic aperture radar (SAR) backscatter have the potential to fill an important gap in the remote monitoring of seasonal snow. Sentinel-1 SAR data have been used previously in an empirical algorithm to generate snow depth products with near-global coverage, sub-weekly temporal resolution, and spatial resolutions on the order of hundreds of meters to 1 km. However, there has been no published independent validation of this algorithm. In this work we develop the first open-source software package that implements this Sentinel-1 snow depth retrieval algorithm as described in the original papers, and evaluate the snow depth retrievals against nine high-resolution lidar snow depth acquisitions collected during the winters of 2019–2020 and 2020–2021 at six study sites across the western United States as part of the NASA SnowEx Mission. Across all sites, we find agreement between the Sentinel-1 snow depth retrievals and the lidar snow depth measurements to be considerably lower than requirements placed for remotely sensed observation of snow depth, with a mean RMSE of 0.92 m and a mean Pearson correlation coefficient $r$ of 0.46. Algorithm performance improves slightly in deeper snowpacks and at higher elevations. We further investigate the underlying Sentinel-1 data for a snow signal through an exploratory analysis of the cross- to co-backscatter ratio ($\sigma_{VH}/\sigma_{VV}$; i.e. cross-ratio) relative to lidar snow depths. We find the cross-ratio increases through the time series for snow depth over $\sim$1.5 m but that the cross-ratio decreases for snow depths less than $\sim$1.5 m. We attribute poor algorithm performance to a) the variable amount of apparent snow depth signal in the S1 cross ratio and b) an algorithm structure that does not adequately convert S1 backscatter signal to snow depth. Our findings provide an open-source framework for future investigations, along with insight into the applicability of C-band SAR for snow depth retrievals and directions for future C-band snow depth retrieval algorithm development. C-band SAR has the potential

to address gaps in radar monitoring of deep snowpacks; however, more research into retrieval algorithms is necessary to better

understand the physical mechanisms and uncertainties of C-band volume scattering-based retrievals.

## 1 Introduction

Runoff from seasonal snow provides water for billions of people (Barnett et al., 2005; Mankin et al., 2015), supplies up to 70 %
of the annual discharge in the western United States (WUS; Li et al. (2017a)), generates clean hydroelectric power, and supports
agricultural and recreation industries at a total value estimated in the trillions of dollars (Sturm et al., 2017). Understanding

the spatial distribution of snow water equivalent (SWE), one of the defining hydrologic variable of the seasonal snowpack, is
essential for effective management of this critical resource (Bales et al., 2006). SWE is the product of snow depth and snow
density relative to water, with snow depth spatial variability providing the majority of the variation in SWE values (Sturm et al.,
2010). Therefore, accurate measurements of snow depth are crucial for global SWE estimation, since measurement of snow
depth is typically much easier and lower cost than direct measurements of SWE.

Current operational snow depth measurement techniques lack either the spatial or temporal resolution necessary to accurately
monitor basin-scale snow depth patterns for a variety of scientific and resource management applications (NASEM, 2018).
Networks of in-situ weather stations (e.g., SNOTEL in the United States) make point measurements of snow depth with high
temporal resolution. However, accurate spatial interpolation required to generate distributed products presents a significant
challenge (Dressler et al., 2006; Bales et al., 2006; Schneider and Molotch, 2016). This challenge is largely due to snow's

typical spatial Wautocorrelation length of 50–200 m (Trujillo et al., 2009). Measurements from spaceborne passive microwave
instruments (Kelly and Chang, 2003; Takala et al., 2011) can be used to produce distributed snow depth products with 12-hour
temporal resolutions. However, passive microwave measurements, at the typically used 37 GHz, saturate in dry snowpacks
approximately 0.8 m deep (Tedesco and Narvekar, 2010; Smith and Bookhagen, 2018), which represents a small fraction of
total snow depth in some regions, and retrievals are unreliable over complex topography (Tong et al., 2010) due to spatial

resolutions at the km to 10s of km scale. No other global operational SWE remote sensing tool currently exists, despite SWE
being one of the largest uncertainties in the hydrologic cycle (National Academies of Science, 2018). Given the challenges
and limitations associated with widely operationalized methods, other techniques are under development to produce spatially-
distributed snow depth and SWE measurements.

High-resolution commercial stereo imagery (Shaw et al., 2020; Hu et al., 2023), airborne lidar (Currier et al., 2019; Deems

et al., 2013) and structure-from-motion (Bühler et al., 2016; Nolan et al., 2015; Miller et al., 2022; Meyer et al., 2022) provide
distributed snow depth maps at meter to submeter-scale spatial resolutions with errors on the order of tens of centimeters (Mc-
Grath et al., 2019; Currier et al., 2019; Deems et al., 2013). The Airborne Snow Observatory (ASO; Painter et al., 2016)
and the Airborne Coastal Observatory (Geospatial, 2021) produce snow depth maps using airborne lidar in mountain basins
across western North America. However, logistical constraints (e.g., cloud cover, tree canopies, platform range, large expense)

typically limit acquisition frequency and spatial coverage. Spaceborne lidar has shown promise for measuring snow depth,
yet currently has high uncertainties (0.5–2 m) in complex terrain and only provides non-repeating sparsely distributed and

infrequent linear transects of point-based returns, requiring high-resolution airborne lidar snow-free surveys to estimate snow depth (Enderlin et al., 2022; Deschamps-Berger et al., 2023; Besso et al., 2024).

Synthetic aperture radar (SAR) is a promising technique to complement new and mature methods for snow depth monitoring. SAR is an active microwave remote sensing that can operate in all weather conditions, does not rely on solar illumination, and is capable of producing datasets at meter-scale spatial resolution from spaceborne platforms. Unlike optical and lidar techniques, SAR signals penetrate the snow surface and interact with the snowpack, allowing for measurements of snowpack properties. The extent of this penetration and which snowpack features are interacted with varies depending on the SAR signal's frequency and polarization (Rosen et al., 2000; Tsai et al., 2019; Marshall et al., 2021). Thus, SAR methods to retrieve snow depth and SWE have the potential to meet the National Academies of Science (2018) Decadal Survey requirement of snow depth and SWE measurements at 100 m spatial resolution.

Numerous techniques have been explored to extract snow depth or SWE from SAR imagery. Such techniques include evaluating backscatter changes to retrieve snow characteristics (Ulaby and Stiles, 1980; Bernier et al., 1999; Shi and Dozier, 2000; Chang et al., 2014; Lievens et al., 2019), using change in travel time information between image acquisitions to approximate SWE changes (Guneriussen et al., 2001; Deeb et al., 2011; Li et al., 2017b; Dagurov et al., 2020; Marshall et al., 2021; Ruiz et al., 2022; Tarricone et al., 2023; Palomaki and Sproles, 2023; Oveisgharan et al., 2023; Hoppinen et al., 2024), exploiting SAR travel time change sensitivity to local slope to capture SWE (Eppler et al., 2022), differencing DEMs for snow depth (Leinss et al., 2018), using differences in the polarimetric response of radar travel times (Leinss et al., 2014, 2016; Voglimacci-Stephanopoli et al., 2021), using travel time changes of frequency subswaths for SWE estimates (Engen et al., 2004), and utilizing phase noise in SAR imagery for snow coverage (Shi et al., 1997; Singh et al., 2008). More detailed reviews of these SAR techniques are available in Tsai et al. (2019), Awasthi and Varade (2021), and Tsang et al. (2022).

However, these SAR-based methods for retrieving snow depth and SWE are all relatively immature and require additional investigation to understand limitations before they can be operationalized. Two recent studies (Lievens et al., 2019, 2022) have demonstrated the potential of deriving spatially distributed snow depth maps on a global-scale from Sentinel-1 (S1) SAR imagery. In the original studies, the technique was validated using snow depth measurements from point-based stations and spatially-distributed modeled data. A recent independent validation effort from Broxton et al. (2024) compared S1 snow depths to ASO lidar-based and University of Arizona (Broxton et al., 2016) modeled depths at 500 m and 1 km spatial resolution. For all S1 pixels, they found moderate coefficient of determination values ($R^2 = 0.62$) and large negative biases ($\sim -50$ %) when compared to the ASO data. However, error metrics improved when flagging for wet snow pixels ($R^2 = 0.89$). Here, we provide another independent validation of the S1 snow depth retrieval technique using spatially-distributed, lidar-based snow depth measurements across multiple sites in the WUS (Abedisi et al., 2022a).

## 1.1 SAR volume scattering snow depth retrieval theory

SAR sensors emit electromagnetic energy in the microwave range (1–300 GHz) and measure the amplitude and phase of the backscattered (returning) waves. In snow-covered terrain the backscattered energy is some combination of returns from vegetation (if present), the snow-air interface, snow volume, ground-snow interface, and ground volume. The exact magnitude

of returning energy from each is a function of the selected radar frequency, incidence angle, vegetation characteristics, snow microstructure, snow liquid water content, snow depth, and the ground surface characteristics (Ulaby et al., 1974; Cihlar and Ulaby, 1974; Naderpour et al., 2022). As a simplification, the features that dominant backscatter are those closest in size to the radar wavelength and interfaces with the largest dielectric changes. For lower frequencies (<≈20 GHz), the main contributors to backscatter are vegetation volumetric backscatter (if present) and (for non-grazing incidence angles) specular reflection from the ground-snow interface, with less significant contributions from the snow-air interface, snow volume, and ground volume scattering. (Long, 1975; Schmugge et al., 1973; Ulaby et al., 1986; Saatchi et al., 1997; Thiel and Schmullius, 2016; Hosseini and Garestier, 2021).

When the radar wavelength is within an order of magnitude of the diameter of snow grains (∼0.1–5 mm), volumetric scattering within the snow volume and at snow-layer interfaces becomes a significant factors in the returning wave amplitudes (Ulaby et al., 1986; Brangers et al., 2023; Tsang et al., 2022). Hence, for SAR frequencies between ≈5–40 GHz, the presence of snow increases volumetric scattering relative to non-snow conditions (Figure 1; Ulaby and Stiles, 1980). Higher frequency SAR systems can exploit this increased volumetric backscatter to retrieve measurements of snow depth and SWE (Tsang et al., 2022). Note that observing this increase in backscatter assumes negligible changes in air-snow interface scattering, vegetation, or ground backscatter contributions. Also, these approaches are generally ineffective in wet snow conditions, where liquid water within the snowpack absorbs microwave energy, leading to marked reductions in backscatter and limiting microwave penetration depth (Stiles and Ulaby, 1980; Tiuri et al., 1984; Bonnell et al., 2021; Lund et al., 2022). In these conditions other SAR techniques such as DEM generation from the wet snow surface may be more appropriate (Leinss et al., 2018).

The relationship between C-band volume scattering and snow depth is an ongoing area of investigation. Initial studies suggested that dry snow has virtually no effect on volumetric scattering at C-band and any mid-winter changes in backscatter were caused by variations in snow-ground interface scattering and variability in the soil dielectric constant (Wegmüller, 1990; Bernier et al., 1999; Sun et al., 2015). However, these studies were limited by shallow (<1 m depth) snowpacks (Bernier and Fortin, 1998; Fuller et al., 2009), solely co-polarized (parallel transmitting and receiving antennas) backscatter (Mätzler, 1987; Fuller et al., 2009; Shi and Dozier, 2000), or an inconsistent ground footprint (Strozzi et al., 1997). These results align with microwave scattering theory as the wavelength at C-band is too large to be scattered by individual snow grains, which are typically <5 mm. Previous studies using tower-mounted radars (Strozzi et al., 1997; Mätzler, 1987) and aerial radar (Bernier and Fortin, 1998) detected either no relationship or even a slight negative correlation between C-band backscatter and snow depth.

Other studies have suggested that dry snowpacks are not fully transparent at C-band. A pair of early studies showed a strong relationship between SWE and the HH backscatter coefficient at 9GHz (Ulaby and Stiles, 1980) and that the depolarization ratio ($\sigma_{HV}^{0}/\sigma_{HH}^{0}$) for 0.28 m of snow depth increased rapidly between 1-8 GHz (Stiles and Ulaby, 1980). Later studies specifically at C-band using artificial snow showed a cross-polarized (orthogonal transmitting and receiving antennas) backscatter increase of 5 dB with a 1 m snow depth increase in a laboratory setting (Kendra, 1995) and then 7 dB increase with a 0.82m snow depth increase in a field setting (Kendra et al., 1998). Two recent tower-based studies showed 2–5 dB increases in co-polarized backscatter for C-band radiation (Naderpour et al., 2022) and significant volume scattering from C-band cross-

polarized backscatter at snowpack layering interfaces (Brangers et al., 2023), likely due to surface roughness effects. More recently, the development of dense media radiative transfer (DMRT) models has suggested that anisotropic clusters of snow grains and multiple scattering effects between interfaces in the snowpack may produce more cross-polarized backscatter from within the snowpack volume at C-band than previous isotropic scattering models suggested (West, 2000; Ding et al., 2010; Chang et al., 2014; Zhu et al., 2023; Picard et al., 2022). The increase in cross-polarized backscatter from these clusters may be sufficiently large to allow for measurements snow depth increases.

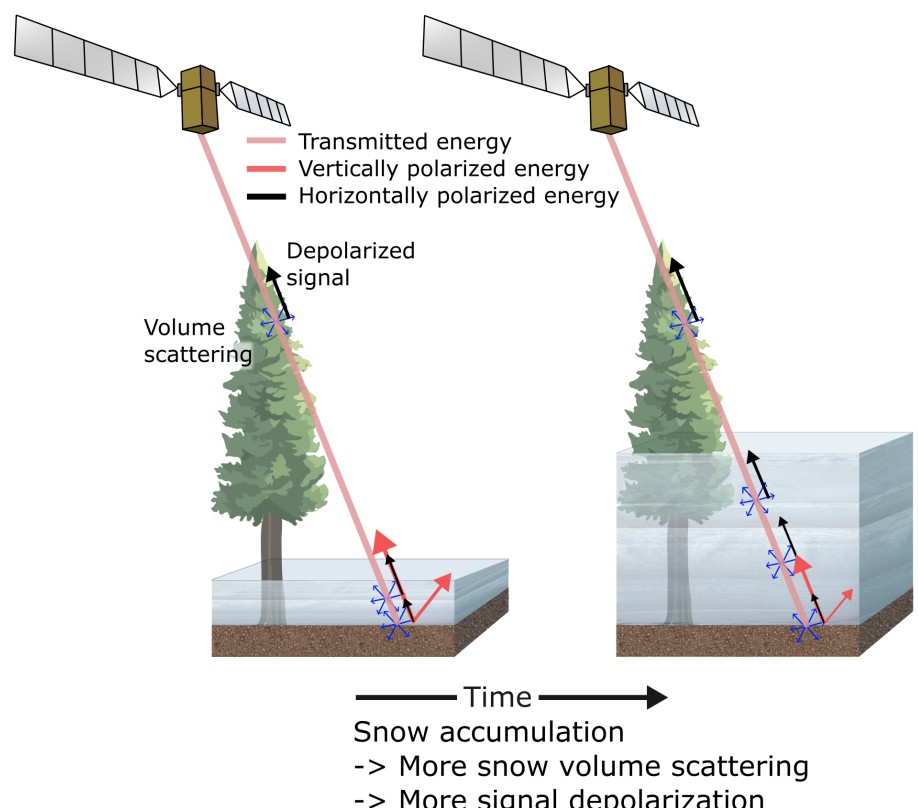

**Figure 1.** Idealized conceptual figure showing the increases in cross-polarized backscatter relative to co-polarized backscatter with increasing snow depth.

## 1.2 Research Objectives

These theoretical results form the basis for satellite-based snow depth retrievals. Lievens et al. (2019) developed an empirical algorithm based on the ratio of VH (cross-polarized) to VV (co-polarized) backscatter, referred to as the cross-ratio, from S1 imagery to map snow depth at 1 km resolution. This approach attempts to reduce the impacts of changes in the soil and geometric signals, which would affect both polarizations, and isolate the snow signal, which is expected to primarily affect the

cross-polarized backscatter. Initial results over the Northern Hemisphere showed mean absolute errors (MAE) of 0.31 m when compared to in-situ station measurements. The technique was further refined in a subsequent study by Lievens et al. (2022) over Switzerland and Austria, where the authors compared the spaceborne retrievals to modeled snow depth changes. The best results were achieved in regions with snow depths greater than 1.5 m, forest cover (FC) less than 80 %, and elevations higher than 1000 m, which would minimize wet snow.

While the results presented by Lievens et al. (2019, 2022) are encouraging, the original works only validated their algorithm against point-based in situ measurements and modeled snow depths. Moreover, a publicly-available version of the algorithm has not been released by the authors, hindering any independent validation and algorithm enhancements. In this study we present an open-source Python package called 'spicy_snow' (Hoppinen et al., 2023) that implements the S1 snow depth retrieval algorithm as described by Lievens et al. (2022). We then evaluate algorithm performance using new spatially distributed lidar snow depth datasets collected during NASA SnowEx 2020–2021 campaigns.

## 2 Methods

### 2.1 Datasets

#### 2.1.1 Sentinel-1 (S1) imagery

The S1 mission is a constellation of polar-orbiting satellites that acquire C-band (5.405 GHz or 5.55 cm) SAR data with a 12-day orbital cycle. We used S1 images acquired in interferometric wide (IW) swath mode, dual-polarized vertical transmit, and vertical/horizontal receive (VV+VH). S1 captures images from the same orbital geometry only every 6, 12, or 18 days. However, due to the overlapping S1 swaths from different orbits most locations see a S1 acquisition every 2—12 days for mid-latitudes and up to daily revisits at polar latitudes. S1 images were processed using the Alaska Satellite Facility's (ASF) HyP3 pipeline (Hogenson et al., 2020) to produce radiometrically terrain corrected $\gamma_0$ backscatter images using GAMMA software (Werner et al., 2000; Wegnüller et al., 2016) and the GLO-30 Copernicus DEM (European Space Agency, 2021). Although this DEM is different from the SRTM DEM used by Lievens et al. (2019, 2022) in their S1 image processing, we selected the GLO-30 dataset in order to avoid inaccuracies inherent in the SRTM data over mountainous regions in North America (Tarricone et al., 2023). Image pre-processing included precise orbit file application, border noise removal, thermal noise removal, radiometric calibration, range-doppler terrain correction, and terrain flattening to produce $\gamma_0$ images at 30 m resolution. We implemented $3 \times 3$ multi-looking processing step to produce images at 90 m resolution, which approximates but does not exactly match the 100 m resolution used by Lievens et al. (2022).

For each study site (Section 2.1.2, Table 1, Figure 2), we downloaded all available (ascending and descending, S1A and S1B) S1 images that contained the bounding box of the lidar validation dataset, beginning on 1 August preceding the winter season. Different relative orbits produce images with changing backscattered power due to variable incidence angles. To account for these differences we normalized the images from each S1 orbit geometry, as done in the Lievens et al. (2022) algorithm. For each specific orbit geometry and polarization, we applied a constant shift to all acquisitions of that orbit geometry's so that

particular orbit geometry time series mean matched the overall mean for that polarization. To correct for outliers, we calculated
the 10th and 90th percentiles of backscattered power for each polarization and subset of images. We then masked any values
that were 3 dB above the 90th percentile or 3 dB below the 10th percentile. We also masked out pixels with local incidence
angles greater than 70° to avoid regions of radar shadow. Additional processing details are given in Appendix A.

### 2.1.2 SnowEx lidar acquisitions

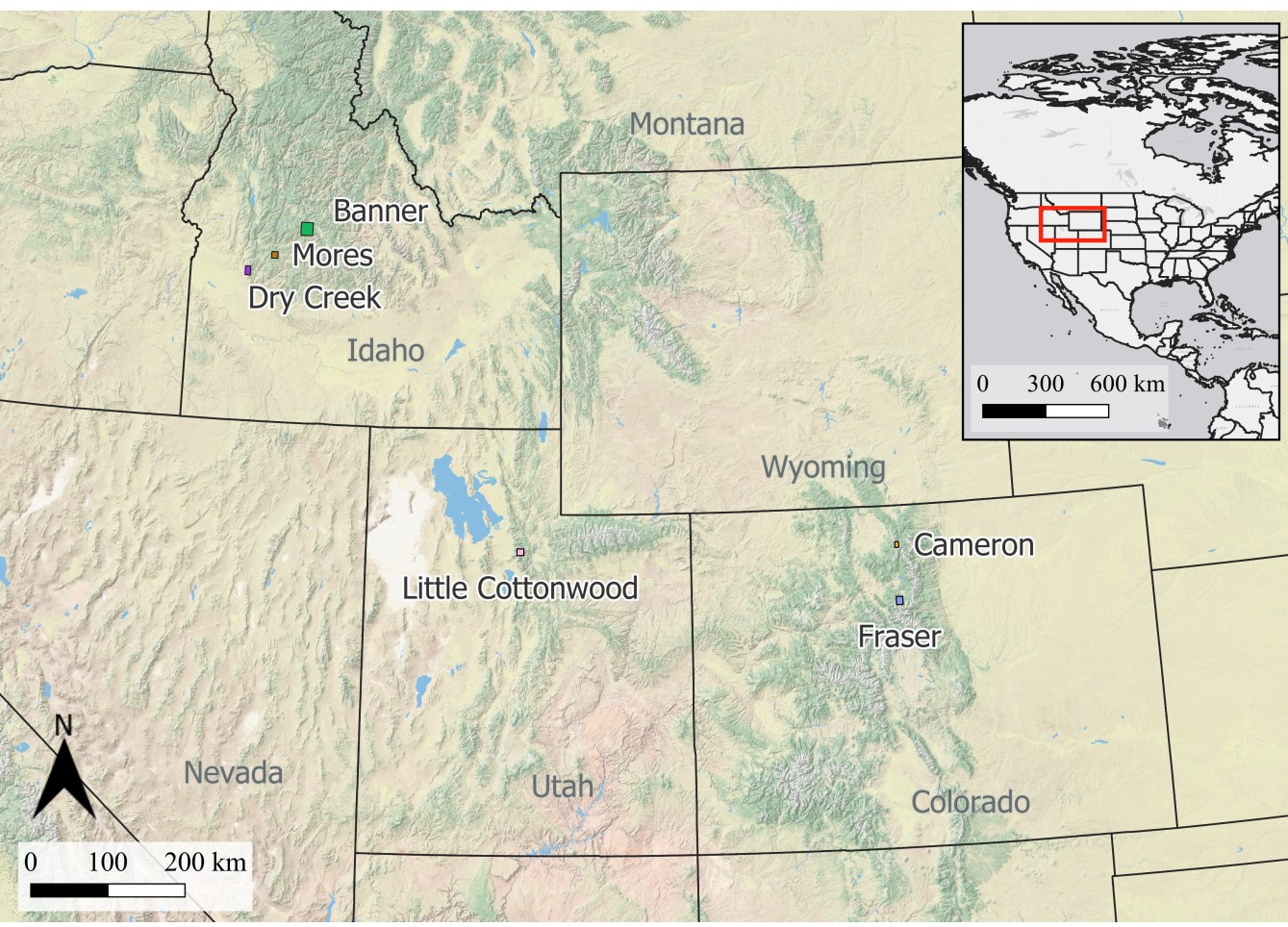

**Figure 2.** Bounding boxes of airborne lidar collected at the NASA SnowEx sites.

The NASA SnowEx campaign (Durand et al., 2019) was a multi-year effort aimed at addressing knowledge gaps in snow
remote sensing and to prepare for a snow-focused satellite mission. During the SnowEx 2020 and 2021 campaigns, Quantum
Spatial Inc. (QSI) acquired snow-free and snow-on lidar validation datasets at six research sites across the WUS (Figure 2):
Fraser Experimental Forest (Fraser) and Cameron Pass (Cameron) in Colorado; Little Cottonwood Canyon (Little Cottonwood)
in Utah; Dry Creek Experimental Watershed (Dry Creek), Mores Creek Summit (Mores), and Banner Summit (Banner) in

**Table 1.** Overview of six study sites and lidar data. Lidar snow depths accuracy's were computed by averaging the 0.5 m resolution lidar data within a 3 m buffer around the SNOTEL location. The snow classes: Montane Forest (MF) Boreal Forest (BF), Prairie (P), and Tundra (T), are defined by Sturm and Liston (2021).

| Site name | Center coordinates | Area (km$^2$) | Elevation range (m) | Snow class | Date(s) (m/d/y) | Accuracy (m) | S1 pixels (count) | SNOTEL | Mean depth (m) |
|---|---|---|---|---|---|---|---|---|---|
| Cameron | −105.890°, 40.538° | 22.1 | 2897–3711 | BF (92 %) MF (4 %) T (4 %) | 3/19/2021 | 0.02 | 2378 | Joe Wright (CO:551) | 1.41 |
| Fraser | −105.894°, 39.885° | 63.2 | 2667–3800 | BF (57 %) MF (40 %) T (3 %) | 2/11/2020 3/19/2021 | 0.03 0.02 | 3847 6787 | Fool Creek (CO:1186) | 1.11 0.86 |
| Little Cottonwood | −111.668°, 40.560° | 28.1 | 1983–3457 | BF (30 %) MF (31 %) T (35 %) P (4 %) | 3/18/2020 | 0.19 | 2827 | Snowbird (UT:766) | 1.81 |
| Banner | −115.184°, 44.268° | 168.7 | 1566–2820 | BF (20 %) MF (40 %) T (20 %) P (20 %) | 2/18/2020 3/15/2021 | 0.02 0.03 | 16415 16692 | Banner Summit (ID:312) | 1.51 1.48 |
| Mores | −115.685°, 43.946° | 34.7 | 1551–2469 | BF (18 %) MF (45 %) T (12 %) P (25 %) | 2/09/2020 3/15/2021 | 0.01 0.06 | 3694 3813 | Mores Creek Summit (ID:637) | 1.79 1.60 |
| Dry Creek | −116.104°, 43.747° | 38.3 | 1233–2279 | MF (97 %) P (3 %) | 2/19/2020 | 0.05 | 3792 | Bogus Basin (ID:978) | 1.05 |

Idaho (Abedisi et al., 2022a, b). Banner, Fraser, and Mores were surveyed in both 2020 and 2021, resulting in nine unique snow depth products (Table 1). QSI processed these data, providing snow-free digital elevation models (DEMs), vegetation height, and (Abedisi et al., 2022a) used these products to produce snow depth maps at 0.5 m spatial resolution. To compare these maps with S1 snow depth retrievals, we aggregated the lidar snow depth measurements at 90 m spatial resolution by taking the average of all 0.5 m lidar snow depth measurements inside each 90 m S1 pixel.

### 2.1.3 Ancillary datasets

The S1 snow depth retrieval algorithm requires FC and snow cover datasets in addition to S1 imagery. Following the procedure outlined in Lievens et al. (2022), we used the Copernicus Global Land Service Proba-V land cover dataset (Buchhorn et al., 2020) at 100 m resolution to quantify FC and mask open-water areas. Additionally, we use the Interactive Multisensor Snow and Ice Mapping System (IMS) (NSIDC, 2008; Helfrich et al., 2007), a daily binary snow cover product at 1 km spatial resolution, to delineate binary snow presence.

## 2.2 Snow depth retrieval algorithm

We implemented a fully reproducible, open-source Python version (Hoppinen et al., 2023) of the S1 algorithm introduced by Lievens et al. (2022). A complete description also appears in Appendix A. The central equation of this pixel-wise approach can be written as:

$$\Delta SD = C\left[(1 - FC) \cdot \Delta(A\gamma_{\mathrm{VH}}^0 - \gamma_{\mathrm{VV}}^0) + B \cdot FC \cdot \Delta\gamma_{\mathrm{VV}}^0\right] \tag{1}$$

where snow depth ($SD$) is obtained within each S1 pixel using the cross-polarized ($\gamma_{\mathrm{VH}}^0$) and co-polarized ($\gamma_{\mathrm{VV}}^0$) S1 backscatter in units of dB, forest cover fraction ($FC$) within the pixel, as well as three empirical tuning parameters ($A$, $B$, and $C$) that are used to control the relative weight of the VH backscatter to VV in the cross-polarized ratio ($A$), the influence of vegetation effects ($B$) and rescale a "snow index" to snow depth ($C$). Note that A and B parameters are dimensionless while C has units of $\mathrm{m\,dB^{-1}}$. Subtraction of cross- and co-polarized backscatter in the logarithmic dB scale equates to a ratio in the linear power scale, and we refer to this $\gamma_{\mathrm{VH}}^0 - \gamma_{\mathrm{VV}}^0$ term as the cross ratio (CR). The $\Delta$ operator in Eq. 1 denotes changes between two S1 images with the same orbital geometry, which may not be the two closest images in time. The S1 algorithm implements Eq. 1 only for pixels with snow present in the IMS data corresponding with the timestamp of the S1 image. Starting with an assumed zero $SD$ on August 1 of a given year, $\Delta SD$ is integrated over time.

The empirical $A$, $B$, and $C$ parameters in Eq. 1 are designed to be tunable to optimize algorithm performance. Lievens et al. (2022) used parameter values $A = 2.0$, $B = 0.5$, $C = 0.44$ optimized to modeled snow depth data over Switzerland. Here, we derived a new set of parameters optimized for the WUS, using the S1 image closest in time to each of the nine lidar acquisitions (Table 1). The time between S1 and lidar snow depth acquisitions was less than two days, except for Mores 2020 (two days, one hour) and Fraser 2020 (five days, 13 hours). As in Lievens et al. (2022), we optimized the $A$ and $B$ parameters by maximizing the Pearson correlation coefficient $R$ and the $C$ parameter by minimizing mean absolute error (MAE) (Webster and Oliver, 2007) between the lidar and algorithm-retrieved snow depths. We varied $A$ between 1 and 3 by increments of 0.1, $B$ between 0 and 1 by increments of 0.1, and $C$ between 0 and 1 by increments of 0.01. Our new WUS-optimized parameter set is $A = 1.5$, $B = 0.1$, and $C = 0.59$, and we used this parameter set in all subsequent analysis. We further investigated the relative impacts of these three tuning parameters on retrieved snow depth, a discussion of which is included in Appendix B.

To ensure that we had effectively implemented the algorithm described by Lievens et al. (2022), we compared the snow depth maps produced for our study sites to corresponding snow depth maps produced as part of the Lievens et al. (2022) effort, known

as C-SNOW. These data are available by request at the C-SNOW data portal (https://ees.kuleuven.be/eng/apps/project-c-snow-data/). Across all study sites, average correlation between our snow depth maps and the Lievens et al. (2022) snow depth maps was 0.64. Differing RTC processing applications (Alaska Satellite Facility's Hyp3 in this study vs. ESA Sentinel Application Platform (SNAP) toolbox used by Lievens et al. (2022)) may partially explain this discrepancy. Additional differences may be explained by updates to the procedure used to generate the C-SNOW data products which are not described in the published article (H. Lievens, Personal Communication, December 25, 2023). These updates include: averaging backscatter changes relative to the previous 6, 12, 18, and 24 days, using the wet snow flags to reduce wet snow-influenced snow depth changes, and different averaging weight vectors for calculating the previous snow index (see Appendix A for a description of the snow index). When compared to lidar snow depth data, we found negligible differences in accuracy between the products produced using our open-source software and C-SNOW. Since the average correlation to the lidar across the nine sites was 0.003 higher for our retrievals relative to the provided data we continued with the open-source retrievals.

## 3 Results

### 3.1 Algorithm performance

Here we assess the performance of the S1 snow depth retrieval algorithm using root mean square error (RMSE) and $R$ to enable a comparison with the results reported by Lievens et al. (2022). Mean site-wide snow depth is variable across the lidar datasets; thus, we also use a normalized RMSE (nRMSE), produced by dividing the RMSE by the site-wide mean snow depth, to enable easier comparison across the sites. For all available measurements across all sites (n = 60,245 pixels), the S1-derived snow depths have an RMSE of 0.92 m (nRMSE = 68 %) and a correlation value of $R = 0.46$ when compared to lidar-derived snow depths (Table 2). For individual study sites, RMSE ranges between 0.65–1.07 m (nRMSE between 57–83 %) with correlation values between $R = 0.02$–0.54. When pixels flagged as wet snow are removed from the comparison, RMSE and $R$ metrics slightly improve at some sites but decline at others (Table 2).

**Table 2.** RMSE and $R$ values for S1 snow depth retrievals as compared to lidar measurements for all pixels (All) and dry snow pixels only (Dry)

| | RMSE (m) | | $R$ | | Bias | |
|---|---|---|---|---|---|---|
| Site | All | Dry | All | Dry | All | Dry |
| All sites combined | 0.92 | 1.03 | 0.46 | 0.45 | -0.49 | -0.02 |
| Banner 2020 | 1.00 | 0.92 | 0.40 | 0.37 | -0.71 | -0.04 |
| Banner 2021 | 0.89 | 1.14 | 0.42 | 0.49 | -0.19 | 0.23 |
| Dry Creek 2020 | 0.74 | 0.78 | 0.21 | 0.24 | -0.43 | -0.43 |
| Fraser 2020 | 0.93 | 1.26 | 0.38 | 0.14 | -0.78 | 0.12 |
| Fraser 2021 | 0.65 | 0.79 | 0.18 | 0.44 | -0.45 | 0.26 |
| Little Cottonwood 2021 | 1.07 | 1.17 | 0.54 | 0.51 | -0.17 | 0.46 |
| Mores 2020 | 1.07 | 0.97 | 0.08 | 0.19 | -0.72 | -0.47 |
| Mores 2021 | 0.91 | 0.91 | 0.40 | 0.34 | -0.51 | -0.15 |
| Cameron 2021 | 1.07 | 1.03 | 0.02 | 0.46 | -0.86 | 0.06 |

Across pixels at all study sites, there is poor agreement between S1-retrieved snow depths and lidar snow depths, particularly where lidar snow depths are less than 1 m (Figure 3a). For individual sites, snow depth distributions broadly fail to match the distributions of snow depth captured with lidar (Figure 3b). At most other sites snow depth is strongly underestimated by the S1 retrieval and retrieved snow depths exhibits much larger dynamic ranges compared to lidar. (Table 2.

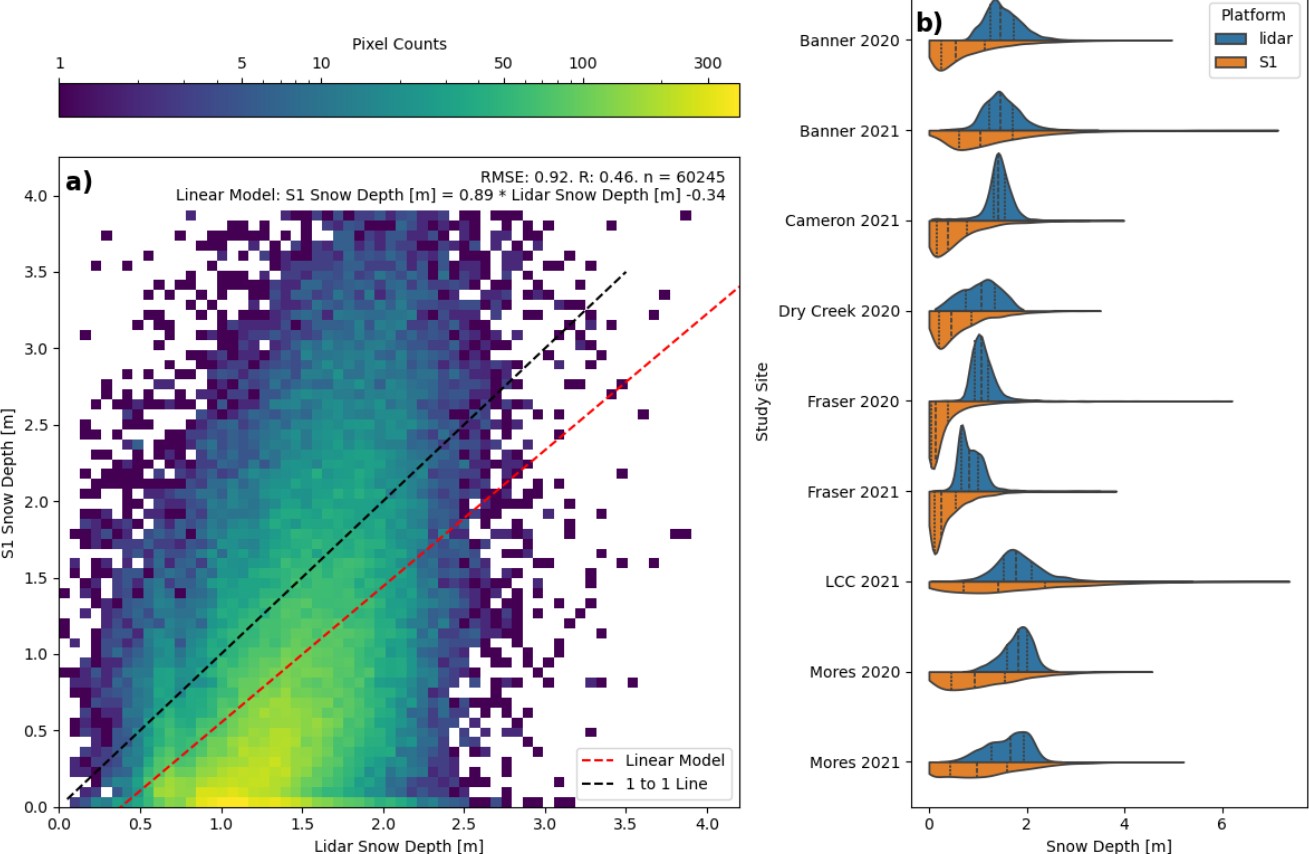

**Figure 3.** a) Site-wise comparison of the distributions of lidar and S1 snow depths. b) pixel-wise log-scaled 2d histogram of lidar vs. S1 snow depths. Dashed lines within histograms represent the 25th, 50th, and 75th percentiles.

We use the Banner 2021 site to qualitatively illustrate the differences between lidar and S1 snow depths (Figure 4). Banner 2021 has a relatively good agreement between S1 and lidar snow depths (RMSE = 0.89, $R$ = 0.42) compared to the other sites. The spatial distribution of snow depth from lidar (Figure 4a) and S1 (Figure 4b) have a first order similarity, with deeper snow depths along the site's central ridge and shallower snow depths at lower elevations to the east and west. However, the S1 algorithm estimates shallower snow depth across considerable portions of the study area (brown in Figure 4c). This negative bias (S1 - Lidar) appears especially prevalent in lower elevation regions (Figure 4d) with higher FC (Figure 4e). Conversely, the algorithm overestimates snow depths in high elevation regions with less tree coverage. Lidar-derived snow depths generally

change smoothly over the landscape, with more abrupt changes in snow depth coinciding with topographic features. In contrast, S1-derived snow depths are noisier, with abrupt snow depth changes that do not coincide spatially with topographic features.

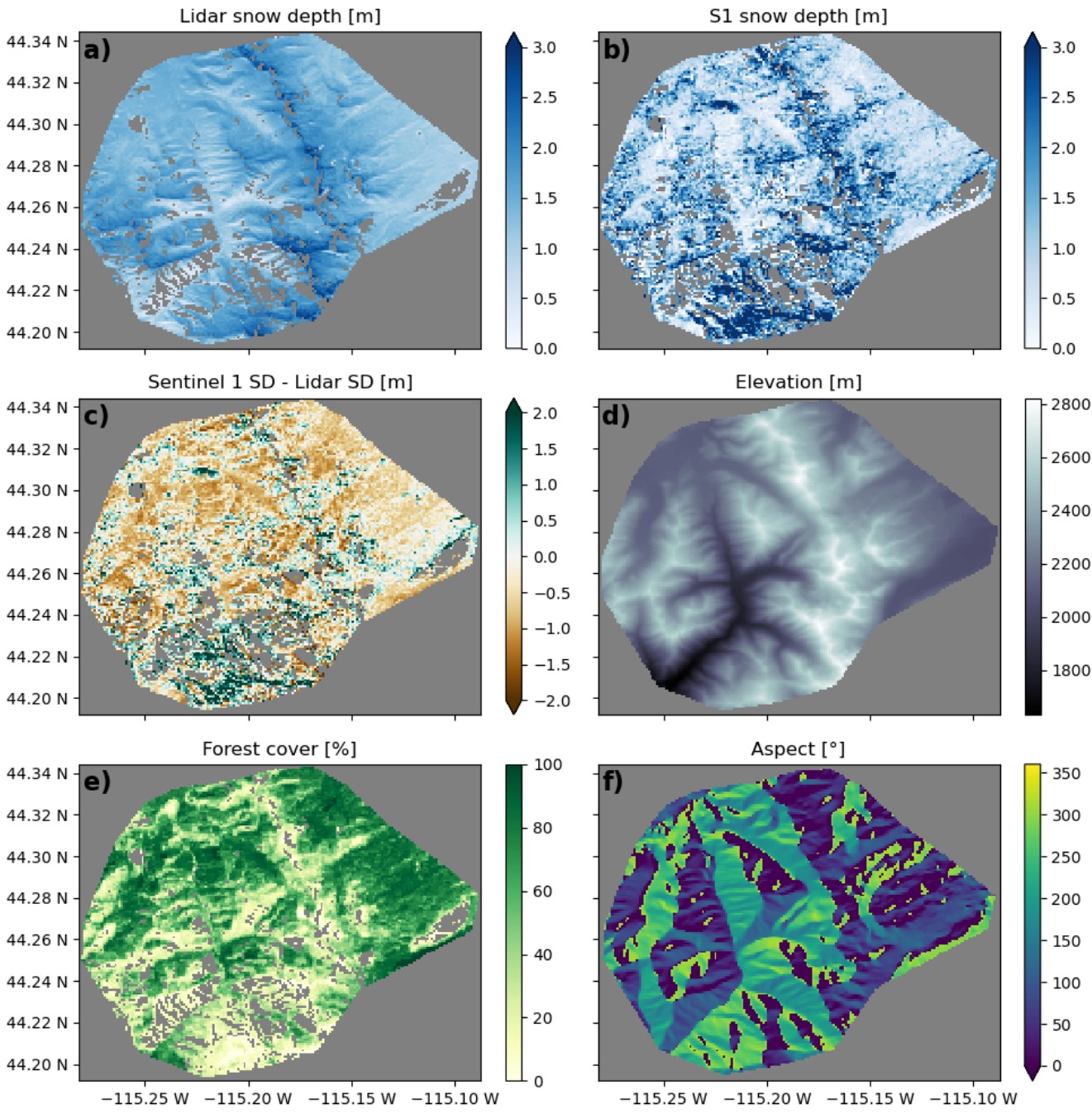

**Figure 4.** From Banner 2021: a) Lidar snow depth, b) S1 snow depth, c) S1 snow depth bias, d) elevation, e) forest cover, and f) aspect angle

We further explore algorithm performance at Banner 2021 within the context of differences in absolute snow depth (measured by lidar), FC, elevation, snow type (dry vs. wet), terrain aspect, and spatial resolution of the datasets (Figure 5). In general, pixels corresponding to a given lidar snow depth bin (e.g., 0–1 m) have a substantially larger range of S1-retrieved snow depths (Figure 5a). Where lidar snow depth is shallower than 2 m, the S1 retrieval algorithm generally underestimates snow depth. Where lidar snow depth exceeds 2 m, the S1 snow depth retrieval mean more closely agrees with the lidar snow depth

mean, but exhibits a considerably wider spread. S1 and lidar snow depths agree best in regions of moderate forest cover (25–75 %), and errors increase in pixels with either very sparse or very dense vegetation, with higher FC leading to underestimated snow depth(Figure 5b). The elevation-dependent results in (Figure 5c) reinforce the spatial patterns visible in Figure 4, with better agreement at higher elevations and underestimated snow depth at lower elevations, although this may also be due to a correlation between elevation and FC at this site. Algorithm performance at Banner 2021 not vary considerably for wet vs.

dry snow (Figure 5d), nor do we observe large variations with respect to terrain aspect (Figure 5e). Lastly, in accordance with Lievens et al. (2022), we find increased agreement between lidar and S1 snow depths at coarser spatial resolutions (Figure 5f).

The impacts of changing snow depth, FC, elevation, aspect, and spatial resolution on retrieved SD accuracy at the other eight sites appear similar to the results shown for Banner 2021 (Figure 6). In general, nRMSE is lowest in regions with deeper snow, moderate FC, and higher elevation. We also note decreasing nRMSE at coarser spatial resolutions across all sites (Figure 6e).

**3.2 S1 cross ratio (CR)**

Despite tuning the S1-retrieval algorithm to lidar-derived snow depths (see Appendix B), snow depths obtained using the algorithm do not agree well with the nine lidar datasets. We consider two possible explanations for this poor agreement, which are not necessarily mutually exclusive. First, the algorithm structure, with its three empirical parameters, is not appropriate for application over the WUS. Second, the underlying S1 data does not provide sufficient information for estimating snow depth

(i.e. there is no S1 snow depth signal). To investigate this second explanation, we compared a timeseries of measured snow depth from the nearest SNOTEL station for each site to a time series of the S1 CR ($\gamma^0_{VH} - \gamma^0_{VV}$) from 1 km buffer around that SNOTEL (Figure 7). A visual comparison reveals a positive correlation between the two variables at most sites (e.g., Banner 2021 and Mores 2020) with little to no relationship at a few sites (e.g., Dry Creek 2020 and Fraser 2020). At some sites, the correlation is weak at the beginning of the accumulation season and becomes stronger as the season progresses (e.g.,

Little Cottonwood 2021).

Separately, we compare the CR signal to lidar snow depth by integrating $\Delta$CR, VH, and VV through time for pixels with IMS snow coverage to the date of the lidar survey at each site (Figure 8). We ran a wilcoxon signed rank test comparing the mean change relative to zero in all bins for $\Delta$CR, $\Delta$VH, and $\Delta$VV (Virtanen et al., 2020). All p-values were below 0.0001 except for the bin from 0-0.5 meters for $\Delta$CR (p = 0.11) and from 0-0.5 meters for $\Delta$VH (p = 0.003). Across all pixels at all

275 sites there is a relationship between the increase throughout the winter in cumulative $\Delta$CR for snow depths at or exceeding 1.5 m and snow depth, but no change or even decrease in $\Delta$CR for snow depths shallower than 1.5 m (Figure 8a). Digging into the two components of $\Delta$CR we find that $\Delta$VH seems to be match the pattern in $\Delta$CR with no change or a decrease throughout the season for snow depths <1.5 meters and increases in backscatter with increasing snow depths above 2 m (Figure 8b). $\Delta$VV

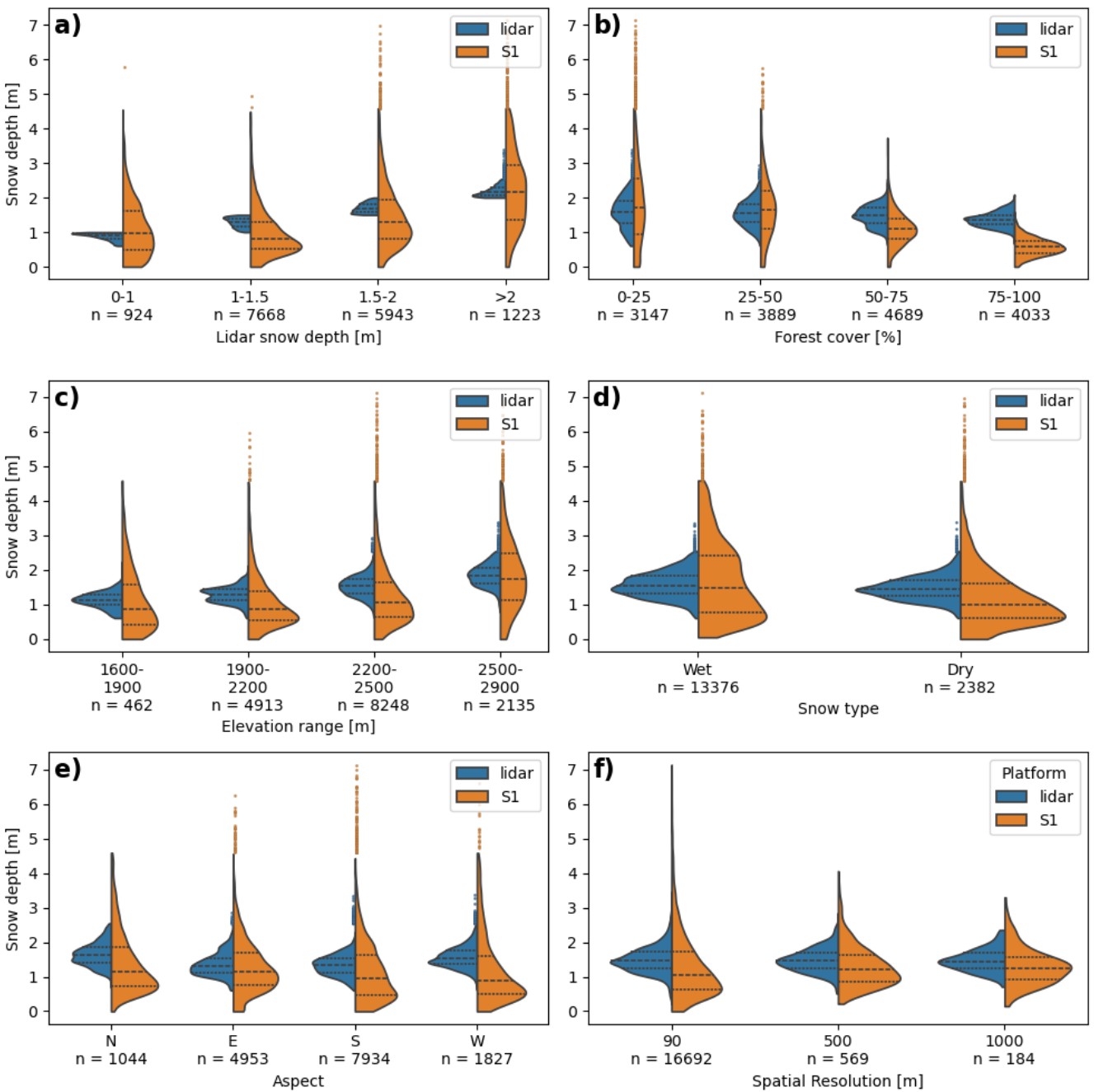

**Figure 5.** Distributions of lidar and retrieved snow depths at Banner 2021 subset by a) snow depth, b) FC, c) elevation, d) wet vs. dry snow, e) aspect, and f) spatial resolution. Values between the 1st–99th percentiles are incorporated into the distributions, while outliers beyond this range are indicated with blue or orange points. Dashed lines indicate the 25th, 50th, and 75th percentiles.

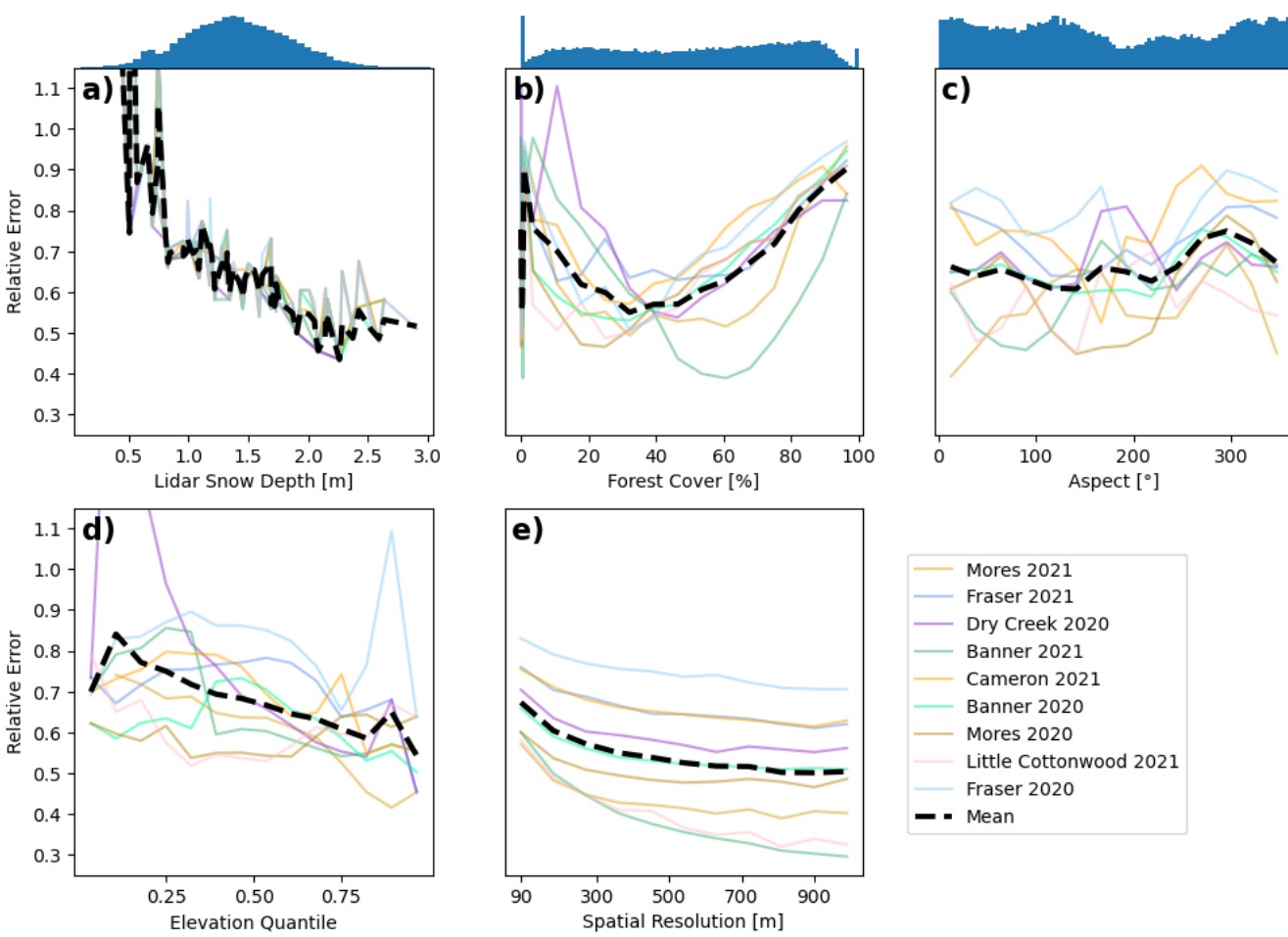

**Figure 6.** Site-by-site nRMSEs along with grouped mean nRMSE for all sites for bins of a) lidar snow depth, b) FC, c) elevation quantile, d) aspect, and e) spatial resolution. Elevation was normalized between 0 and 1 at each site to improve comparison of the intra-site trends. Note that a-c have histograms of their distributions shown above them. Since the distributions were even for the elevation quantiles (d) and spatial resolutions e) they are omitted.

shows a different pattern with snow-on increases in $\Delta$VV for bins below 2 meters and then decreasing $\Delta$VV for bins above 2 meters (Figure 8c).

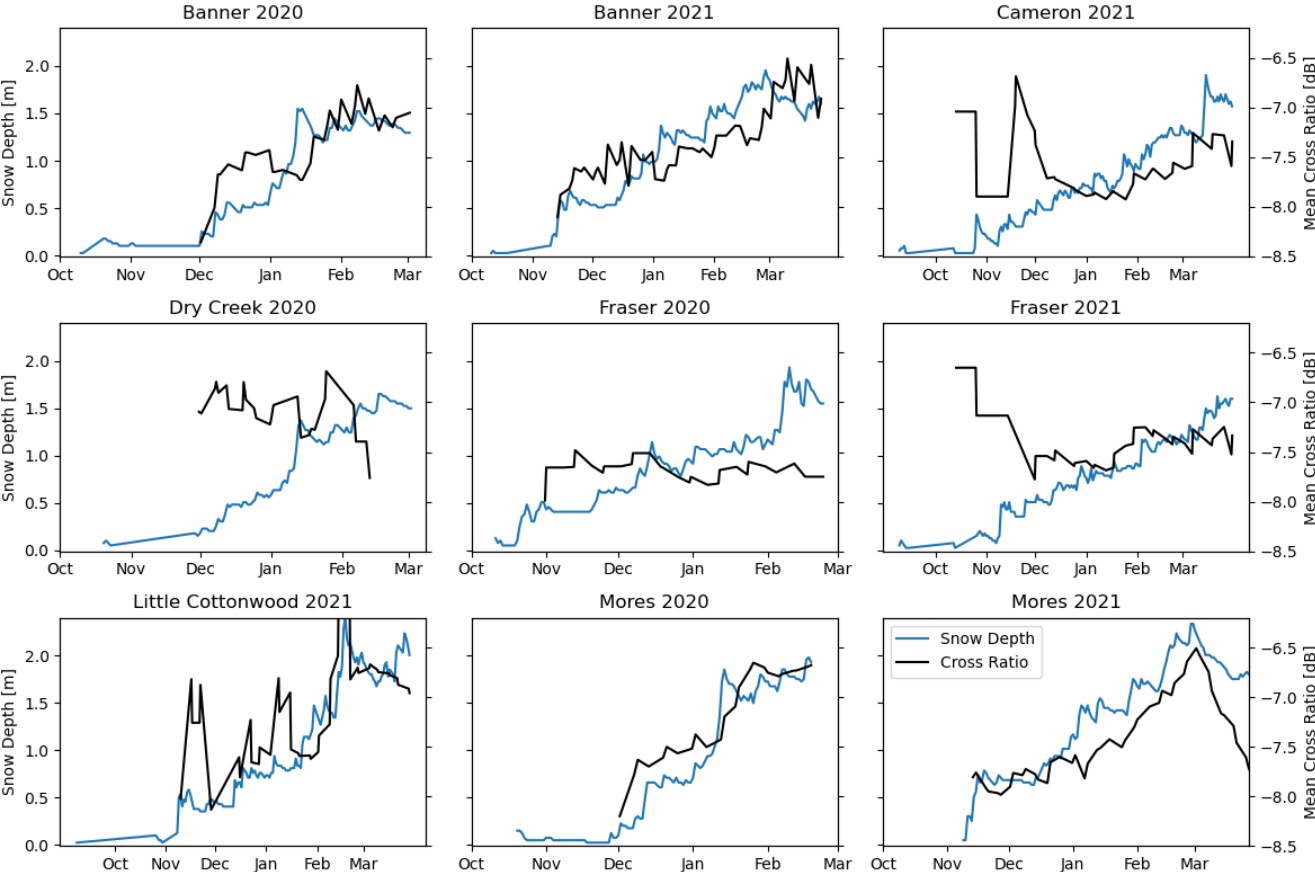

**Figure 7.** Timeseries of mean S1 cross ratio for a 1-km buffer around the SNOTEL ($\gamma^0_{VH} - \gamma^0_{VV}$, blue lines) and measured snow depth from the SNOTEL site (black lines) for all sites. Note that the length of the timeseries varies between sites.

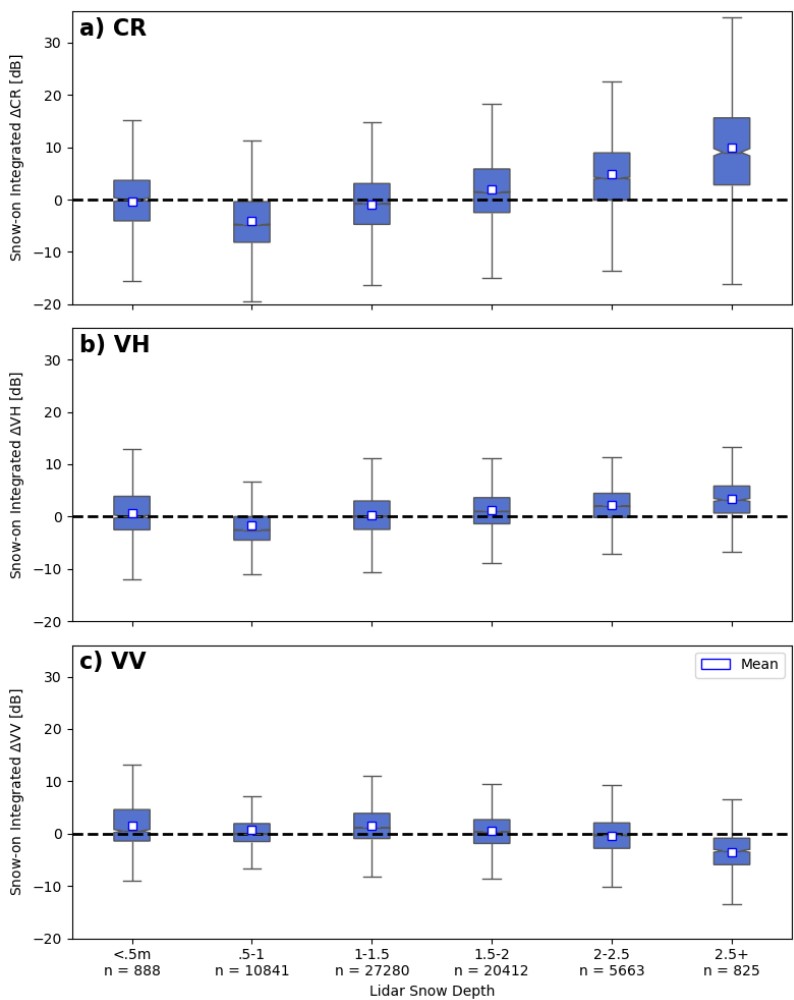

**Figure 8.** Time-integrated change for snow-on periods in a) cross ratio (ΔCR) b) ΔVH c) ΔVV for different snow depths across all pixels and sites.

## 4 Discussion

### 4.1 Snow depth retrieval accuracy

Across all pixels and study sites, we find that the S1 backscatter algorithm proposed by Lievens et al. (2022) captures some of the snow depth spatial distribution ($R = 0.46$; Table 2), but struggles with estimating appropriate magnitudes of snow depth (RMSE= 0.92 m, nRMSE= 68%). Three sites have correlations below 0.2, four sites have RMSE values $> 1$ m, and all sites have nRMSE values greater than 50 % of their site-wide mean snow depths. Only one site (Cottonwood 2021) has a correlation coefficient exceeding 0.5 and one site (Frasier 2021) has an RMSE lower than 0.7 m. Unfavorable sites, such as Cameron 2021 and Mores 2020, have $R$ values as low as 0.02 and 0.08, respectively. While errors improve at coarser spatial resolutions (Figure 6e), nRMSE values range between 31–74 % across all sites, even at the coarsest 1000 m resolution. At sites with deeper snow depths (e.g. Banner 2021, Little Cottonwood 2021), the S1 retrieval algorithm appears to capture first-order spatial patterns in snow depth, despite meter-scale RMSEs.

To better understand the algorithm application, we explored the effects of various environmental and geophysical variables on S1 snow depth retrieval accuracy (Figures 5 and 6). For all sites, nRMSE decreases with increasing snow depth. This improved performance in deeper snow is expected due to increased depolarization and correspondingly higher signal-to-noise ratio (SNR; increased snow depth change related signal relative to other sources, such as thermal noise or radar speckle). When we compare S1 and lidar-derived snow depth in dry snow across all sites (Figure 3a), we find little to no correlation below lidar snow depths of approximately 1.5 m ($R = 0.02$ for 0 - 1 m), with an improved correlation for deeper snow ($R = 0.29$ for 2 m+). We also note decreasing nRMSE with increasing elevation, though we expect considerable correlation between snow depth and elevation across our sites.

At C-band wavelengths, SAR signals within the snowpack interact primarily with large anisotropic grains, grain clusters and at layer interfaces within the snowpack rather than individual snow grains (Naderpour et al., 2022; Tsang et al., 2022; Brangers et al., 2023). While in most cases snow depth is likely correlated with volume scattering from snow layers, other factors controlling the snowpack's structural characteristics additionally impact volume scattering. Spatiotemporal variability of snowpack structure (i.e. faceted grains, ice layers) that is not correlated with snow depth is an important source of uncertainty that may contribute to the poor overall performance of the snow depth retrieval algorithm. Additionally, surface-scattering contributions throughout the season that are uncorrelated with snow depth change could also impact the CR time series leading to uncertainties in retrievals.

The effects of FC on algorithm performance are complicated (Figure 6b), with high errors occurring in areas with dense forest cover (i.e., FC $> 75\%$). Dense vegetation cover is typically associated with elevated levels of SAR volume scattering (Vreugdenhil et al., 2020). As such, a strong vegetation volume scattering signal may overwhelm a weaker signal due to increasing snow depth. Indeed, nRMSE values decrease with decreasing FC down to approximately 35 %. However, errors increase again when FC drops below 30 %. This decline in retrieval performance for sparse tree coverage is unexpected, as 1) previous research found performance improvements with decreasing forest cover (Lievens et al., 2022) and 2) decreasing volume scattering from non-snow sources is expected to improve the snow-related SNR. This observed decline in performance

is potentially caused by very deep snow pixels at high elevations where forest cover is sparse. All deep snow outliers ($> 3$ m) are located where FC is low, most with being below 10 % FC. The retrieval algorithm is optimized for mean snow depths ($\sim 1$–2 m), so algorithm performance likely degrades in extreme snow depth cases. As such, we interpret poor retrieval algorithm performance for low FC values to be caused by algorithm design rather than low SNR.

Terrain aspect influences snow deposition and melting, with east-facing slopes receiving more wind-deposited snow in the WUS, and south and west-facing slopes receiving more direct solar radiation or during warmer periods of the day in the Northern hemisphere (Mock and Birkeland, 2000). Therefore we might expect degraded performance on south aspects (wet snow) and improved performance on east aspects (deeper snow), but average nRMSE values across all sites does not vary substantially across aspect angles (Figure 6d). nRMSE curves for individual sites vary drastically in shape, with differences likely caused by complex interactions between satellite incidence angle, FC, and wet snow effects. The specific impacts of aspect on algorithm performance are still unclear and a potential investigation for future work.

Recent work by Broxton et al. (2024) showed a marked performance increase when excluding wet snow flagged pixels. Therefore, algorithm performance was expected to decline substantially for wet snow-flagged pixels. In wet snow, maximum penetration depth of incident C-band radiation is on the order of 10 cm (Casey et al., 2016), which attenuates the signal and prevents the volume scattering that occurs in dry snow. Unexpectedly, we found that algorithm performance did not improve across all sites when pixels flagged as wet snow were masked (Table 2, Figure 5d). The results from Table 2 suggest that the wet snow flag in the algorithm is not correctly separating wet snow from dry or refrozen snow. Additionally, there may be non-snow scattering mechanisms in shallow snowpacks that trigger the wet snow flag when the snow present is actually dry. For more details on wet snow considerations, see Appendix B2.

We finally consider the effect of spatial resolution on algorithm performance. We find that at 90 m spatial resolution, S1 snow depth retrievals have meter-scale RMSEs, suggesting limited utility for accurately capturing snow depth at this resolution across our WUS domain. Coarser resolution S1 retrievals show better agreement (nRMSE = 0.50, r = 0.47 at 1km) with similarly-resampled lidar measurements (Figure 6e), suggesting that the algorithm is better suited for providing large-scale information about snow patterns that may be valuable for water resource managers and hydrologic modeling. As horizontal resolution coarsens, nRMSE decreases up to 300–500 m, after which improvements level off. This improvement at coarser resolutions may be related to the relatively subtle C-band snow volume scattering signal compared to background noise from SAR speckle, variations in ground and vegetation properties, and other sources.

However, we note that these spatial resolution results need to be interpreted carefully, as 1) spatial averaging decreases the standard deviation and sample size of the S1 snow depth distribution, which can artificially decrease RMSE, and 2) algorithm parameter fits may be improved as the lidar and S1 snow depth distributions are spatially averaged. "Snow Index" values from the S1 retrieval algorithm are converted to snow depth by scaling with the single $C$ parameter, which was optimized by minimizing MAE between the S1 and lidar snow depth data across all sites. Meanwhile, spatial averaging brings individual pixel values closer to the mean values of the distributions, oversimplifying important spatial patterns of snow depth distribution but improving the fit of the $C$ parameter to the S1-derived Snow Index and lidar snow depth data. While simplification of the data via spatial averaging may improve the model fit to the data, this does not necessarily indicate underlying correlation

between the two datasets, only that two simple surfaces can be more easily fit together than two complex surfaces using a single empirical parameter. Thus, improvements in algorithm performance with decreasing spatial resolution must be interpreted as a potential artifact of the particular error metric and algorithm structure.

    Overall, these results suggest that S1 snow retrievals agree best with lidar snow depth measurements in regions with snow-packs deeper than 1.5 m, moderate FC ($\sim$35%), and spatial resolutions between 500–1000 m, a set of conditions generally in

accordance with Lievens et al. (2022). Even under these ideal conditions, nRMSE values for all sites exceed 40 %, well above the 10 % target at 100 m spatial resolution set in the National Academy of Sciences 2017–2027 Decadal Survey (NASEM, 2018). Nonetheless, no current sattlited-based operational snow depth product exists. Even the proposed Ku-and X-band missions from NASA and the Canadian Space Agency will saturate at $\sim$1–2 m snow depth Tsang et al. (2022). Continued exploration to improve this technique is warranted, especially focused on areas of deeper snow ($>$1.5 m).

## 4.2   Cross ratio time series

To provide context for our evaluation of S1 snow depth retrieval accuracy, we performed a brief exploration of the CR time series for each of our study sites. We found that CR appears to be correlated with SNOTEL snow depth at some sites (e.g. Banner 2021, Little Cottonwood 2021, and Mores 2020 and 2021) correlated only in mid-late winter at other sites (e.g. Banner 2020, Cameron 2021, and Fraser 2021), and uncorrelated at others (e.g. Dry Creek 2020 and Fraser 2020). We note that sites

with deeper snow depths tended to exhibit the strongest qualitative correlation between CR and snow depths. These results suggest that while there likely is snow depth information in the S1 CR, there is large spatial and temporal variability in the snow depth SNR. Ideally, S1 retrieval algorithms should only be applied where snow depth signal is detectable. Even with additional sources of snow depth information, identifying these periods in real-time will be a challenge and time-series analysis may be required along with ancillary modeled, higher-frequency SAR, or optical datasets.

We further compared the spatial mean of time-integrated $\Delta$CR, $\Delta$VH, $\Delta$VV with lidar snow depth data. We found that snow depths below 1.5 m do not appear to cause a detectable increase in S1 depolarization over the time leading up to the lidar acquisition at our study sites, which likely explains poor retrieval algorithm performance in shallow snow and contribute to poor early-season performance of the algorithm at some study sites. The exact reason for this change or decrease in $\Delta$CR below 1.5 m is unclear and should be a subject of future research, but some potential explanations are: 1) wet snow at these shallow,

potentially lower elevation, sites prevent signal penetration and depolarization, 2) early season faceting in these areas leads to large increases in depolarization early in the season that decrease as snow depth increases and faceted grains are rounded 3) simply not enough anisotropic grains, grain clusters, and layers for increases in $\Delta$CR to be captured. Comparing the $\Delta$VV and $\Delta$VH components we find that VV is relatively unchanged to even slightly increased for snow depth bins <2 m but we see decreases above 2 m of snow depth. The especially large decrease for 2.5 m+ (-3.55 dB) is surprising for VV. A possibility for

this decrease is that these especially deep snow depths may be capable of attenuating the ground signal a considerable amount even in the VV polarization. The $\Delta$ VH changes generally matched the trends of $\Delta$ CR and suggest we have some unknown factor decreasing VH backscatter from 0.5-1 m of snow depth. Also that for snow depths above 1.5 m we see significant increases in backscatter with snow depth but that the spread will be quite large until snow depths above 2 m.

These results, when combined with previous research on C-band depolarization, suggest that S1 CR data may be a potential source of information in deep snow (>1.5 m), but that snow depth retrieval algorithms using only S1 data will not be reliable until snow depths reach a threshold close to 1.5 m. Where maximum snow depths are shallower, or snow is wet, the S1 CR may not provide useful snow depth information.

### 4.3 Limitations and future work

Interpretation of algorithm performance is complicated by a poor understanding of the underlying physical processes and scattering mechanisms that affect the CR. If the time-integrated CR signal contains information related to changing snow depth at the surface, the snow-related effects are subtle and difficult to untangle in shallow snowpacks (Figures 7 and 8). However, the S1 algorithm we evaluated has no method to indicate where snow depths have surpassed this minimum snow depth threshold, and instead requires in situ data to identify those regions where future changes in the CR signal may be related to changes in snow depth. This is a significant challenge for global application of the algorithm, where vast snow-covered regions do not have in situ data available for reference. Future algorithm development should integrate additional data sources (e.g., passive microwave satellite data, future higher frequency SAR approaches, interferometric SAR approaches, polarimetric SAR approches, or physically-based snow accumulation models) to derive snow depth changes early in the accumulation season.

While incremental improvements to the algorithm may still be possible with additional analysis, parameter tuning, or improved ancillary datasets, perhaps more important is a better understanding of the physical mechanisms controlling the CR signal. This is beyond the scope of a single study and will likely require an iterative approach that considers modeling efforts, laboratory or small-scale field studies, and satellite data. In tandem with future investigations into the CR signal, we advocate for the development of novel approaches for harnessing the snow information that may be present in C-band SAR data. More effective algorithms could incorporate results from radiative transfer models, which would allow for more detailed explorations of potentially covariated scattering mechanisms related to vegetation, snow wetness, and soil properties (Zhu et al., 2023; Borah et al., 2024) Alternatively, machine learning approaches, including physics-informed neural networks, may result in more accurate snow depth retrievals from S1 data and provide insights that guide subsequent modeling and field studies. Lastly, the algorithm presented here has known differences when compared to Lievens et al. (2022) due to the closed-source nature of their work. Until the original code is released in an open-source framework, additional development and improvements from the larger snow remote sensing community will be hindered.

Finally a few other suggestions for future work on backscatter based snow depth retrievals at C-band: (1) analyzing speckle-related uncertainty by pixel resolution to better understand the trade-offs between resolution and uncertainty for this retrieval technique, (2) assessing the impacts of incidence angle on the depolarization and amplitude relationship to snow depth, (3) expanding our SNOTEL analysis of CR to include a larger set of SNOTEL stations.

## 5    Conclusions

In this study we present an independent evaluation of a promising S1 volume scattering-based snow depth retrieval algorithm proposed by Lievens et al. (2022). We developed an open-source Python package implementing a version of the algorithm (Hoppinen et al., 2023) and compared S1 snow depth retrieval algorithm results to nine mid-winter lidar snow depth retrievals over the WUS collected for the NASA SnowEx campaign. Over all study sites, we find that S1 snow depths agree poorly with lidar snow depths, with a mean RMSE of 0.92 m and a mean correlation of 0.46. We find moderate improvements in algorithm

performance in deeper (>1.5 m) snow and FC around 35 %; however, even under these ideal conditions mean nRMSE is 40 %, above the 10 % target at 100 m spatial resolution set in the National Academy of Sciences 2017–2027 Decadal Survey (NASEM, 2018).

To help explain algorithm performance, we briefly explore the S1 CR time series data that the algorithm relies on. We find that the S1 CR is visually correlated with snow depth at some sites, though this correlation sometimes only begins in mid-late

winter. We find a relationship between snow depth and CR signal above ∼1.5 m but no detectable positive time-integrated change in S1 CR for snow depths less than ∼1.5 m. We therefore attribute poor algorithm performance partly to lack of information in the CR when the snow SNR is very low, and partly due to algorithm structure, which fails to reliably convert change in S1 CR to snow depth where SNR is high.

Given the inconsistent nature of the snow depth signal in S1 CR data, we recommend that algorithms using these data

integrate other sources of snow depth information to identify conditions where S1 data are likely to be useful. Future efforts would benefit from improved understanding of the physical mechanisms controlling the interaction between spaceborne C-band radar measurements and the snow-covered landscape. At the same time, more complex empirical algorithms or machine learning approaches may be able to more accurately translate changes in S1 backscatter to snow depth. Measuring global snow depth and SWE from space will require a synergistic approach including various remote sensing techniques, modeling

approaches, and in situ data sources. While questions remain how to best utilize S1 for snow depth, we believe C-band SAR remote sensing products will be a valuable tool in monitoring global snowpacks.

*Code availability.*    The repository for running Sentinel-1 snow depth retrievals using this algorithm is available at: github.com/SnowEx/spicy-snow. Analysis and figure creation code is available at: github.com/ZachHoppinen/spicy-analysis

## Appendix A:  Sentinel-1 Snow Depth Retrieval Algorithm Details

The retrieval algorithm relies on the assumption that no snow exists on the surface at the beginning of the timeseries (we use August 1st for the Northern Hemisphere). Snow depth is calculated iteratively by attributing increases in backscatter to increases in snow depth. The IMS snow presence dataset) (NSIDC, 2008) is incorporated to avoid misattributing backscatter changes from other non-snow factors. Snow depth at each pixel is set to zero until the IMS dataset indicates snow presence, and snow depth is also set to zero after melt-out.

The primary S1 input to the snow depth retrieval algorithm is the cross-ratio of co- and cross-polarized backscatter. The cross-ratio is calculated at every valid pixel ($i$) over all available image acquisitions ($t$) by taking the ratio of VH to VV backscatter in a linear scale, or equivalently by subtracting VH from VV in a logarithmic [dB] scale:

$$\gamma^0_{CR}(i,t) = A\gamma^0_{\text{VH}}(i,t) - \gamma^0_{\text{VV}}(i,t) \tag{A1}$$

where $A$ is an empirical fitting parameter used to control the relative weight of the VH backscatter to VV.

Next, two backscatter change variables are calculated between the image at the current timestep ($t$) and the prior timestep ($t_{\text{pri}}$). Depending on the study site and orbit geometries, the time elapsed between $t$ and $t_{\text{pri}}$ can be 6, 12, 18, or 24 days. The change in the cross-polarized to co-polarized backscatter ratio ($\Delta\gamma^0_{\text{CR}}$) is given by

$$\Delta\gamma^0_{\text{CR}}(i,t) = \gamma^0_{\text{CR}}(i,t) - \gamma^0_{\text{CR}}(i,t_{\text{pri}}) \tag{A2}$$

and the change in the co-polarized backscatter ($\Delta\gamma^0_{\text{VV}}$) is given by

$$\Delta\gamma^0_{\text{VV}}(i,t) = \gamma^0_{\text{VV}}(i,t) - \gamma^0_{\text{VV}}(i,t_{\text{pri}}) \tag{A3}$$

Vegetation causes significant cross-polarized backscatter that may obscure the snow-depth related signal. Consequently, a weighted combination of $\Delta\gamma^0_{\text{VV}}$ and $\Delta\gamma^0_{\text{CR}}$ is implemented using the forest cover fraction ($FC$, bounded between 0 and 1):

$$\Delta\gamma^0(i,t) = (1 - FC(i)) \cdot \Delta\gamma^0_{\text{CR}}(i,t) + B \cdot FC(i) \cdot \Delta\gamma^0_{\text{VV}}(i,t)q \tag{A4}$$

This weighted combination is parameterized by the second empirical fitting parameter $B$ that controls the relative influence
of the co-polarized backscatter change on the final snow depth retrievals. To remove outliers, we masked pixels in the result of (A4) with backscatter changes more than $+3$ dB and less than $-3$ dB.

A snow change index ($SI$, units of dB) captures changes in $\Delta\gamma^0$ over time, taking in information from multiple previous snow indexes and snow coverage data from the IMS. The algorithm is initiated with $SI$ set to 0 for all pixels, and $SI = 0$ as long as the IMS dataset indicates no snow presence. Once snow presence is indicated, a previous snow index is calculated that
takes the weighted average of the snow indexes centered around the last time step from the same orbital geometry (6 or 12 days ago) combined with the snow indexes from around that previous time step (+- 5 days or +- 11 days) (Equation A5) with weights that are the inverse distance in days between the previous time step and that image's acquisition date (Equation A6).

$$SI(i,t_{pri}) = \frac{1}{w} \sum_{t_{image}=t_{pri}-RI+1}^{t_{pri}+RI-1} w \times SI(i,t_{image}); RI \in 6, 12, 18, 24[\text{days}] \tag{A5}$$

and $w$ given by:

$$w = \begin{cases} [1..6..1] & RI = 6 \\ [1..12..1] & RI = 12 \\ [1..24..1] & RI = 24 \end{cases} \tag{A6}$$

For example, an image captured on January 30th in an orbital geometry that captures an image every 6 days ($RI = 6$) would multiply all the previously calculated snow indexes from January 19th to 29th (January 24th $\pm$ 5 days) by the repeat interval minus number of days separating each images from the previous image acquisition date (January 24th) so a vector of [1, 2, 3, 4, 5, 6, 5, 4, 3, 2, 1]. This sum would then be divided by that same vector with days without images removed to get the previous snow index.

The current time step's $\Delta\gamma^0$ is then added to this previous snow index to calculate the current snow index. If the currently calculated snow index is negative it is set to zero for this time step (Equation A7).

$$SI(i,t) = \begin{cases} \max(0, [SI(i,t_{pri}) + \Delta\gamma^0]) & IMS = \text{Snow} \\ 0 & IMS = \text{No Snow} \end{cases} \tag{A7}$$

Next, we convert the current snow index in dB to snow depth in meters by multiplying it by the parameter $C$ (Equation A8). $C$ controls the increase of snow depth correlated with increasing backscatter and was varied between 0 to 1 in increments of 0.01.

$$SD(i,t) = C * SI(i,t) \tag{A8}$$

Finally, a binary wet snow flag is applied with the intention to identify changes in backscatter due to wetting of the snow (causing a strong decrease) or refreezing (causing a strong increase) instead of changes in snow depth. Since different orbit geometries have different local incidence angles and acquire data at different times of the day (06:00 and 18:00 LT), the wet snow flag is only calculated for changes between images of the same orbital geometry. Additionally, once a pixel has been flagged as wet it continues to be wet until a refreezing event occurs at that pixel.

A pixel is flagged as wet if the cross polarized ratio drops by more than 2 dB (wet snow threshold) from the previous image with the same orbital geometry for a pixel with less than 50% FC, or if the co-polarized backscatter drops by more than 2 dB for pixels with greater than 50% FC. This wet snow flag persists for that orbital geometry until an increase of 1 dB (freeze snow threshold) in the cross-polarized ratio (for regions of FC $< 50\%$) or co-polarized ratio (FC $> 50\%$) is observed, after which point the pixel is flagged as dry until the next drop in backscatter is observed (Equation A9).

An "alternate wet snow flag" is also applied if $SI$ drops below zero in a region where the IMS still indicates snow presence, which attempts to capture snow wetness in regions of shallow or patchy snow cover or highly vegetated areas.

$$\text{Wet Flag}(i,t) = \begin{cases} \text{wet} & \Delta\gamma^0_{CR/VV}(i,t) < -2 \text{ dB} \\ \text{wet} & \text{Wet Flag}(i,t_{pri}) = \text{wet}; \Delta\gamma^0_{CR/VV}(i,t) < +1\text{dB} \\ \text{wet} & SI(i,t) < 0; IMS(i,t) = \text{Snow} \\ \text{dry} & t = \text{August 1st} \\ \text{dry} & \Delta\gamma^0_{CR/VV}(i,t) > +1 \text{ dB} \\ \text{dry} & \text{Wet Flag}(i,t_{pri}) = \text{dry}; \Delta\gamma^0_{CR/VV}(i,t) > -2 \text{ dB} \end{cases} \tag{A9}$$

After February 1st, if a pixel was flagged as wet for 50% or more of the previous 4 observations from the same orbital geometry, we consider the snowpack to be permanently wet at that location and flag as wet the remainder of the time series until the next August 1st.

## Appendix B: Parameter Optimization

### B1 Parameter Importance

The optimal parameter values for our WUS validation dataset are $A = 1.5$, $B = 0.1$, and $C = 0.59$. Of these three parameters, changing $C$ has the largest impact on RMSE ($\frac{\partial \text{RMSE}}{\partial A}$=0.207m, $\frac{\partial \text{RMSE}}{\partial B}$=0.176m, $\frac{\partial \text{RMSE}}{\partial C}$=0.908m/(m/dB), Table B1). Because $C$ is used as a scaling parameter in (1), it has no impact on $R$. Modifying $B$ has a larger impact on scene-wide $R$ values than does changing $A$ ($\frac{\partial \text{R}}{\partial A}$ = 0.035, $\frac{\partial \text{R}}{\partial B}$ = 0.101). However, when considering only pixels with FC < 25%, changing $A$ has a larger impact on $R$ and RMSE. In contrast, $B$ increases in importance for high FC pixels.

| Table B1: Parameter Sensitivity | | | |
|---|---|---|---|
| | **All Pixels** | **<25% FC** | **>75% FC** |
| $\partial$ RMSE / $\partial$ A [m] | 0.207 | 0.454 | 0.144 |
| $\partial$ RMSE / $\partial$ B [m] | 0.176 | 0.019 | 0.367 |
| $\partial$ RMSE / $\partial$ C [m/(m/dB)] | 0.908 | 1.871 | 0.412 |
| $\partial$ R / $\partial$ A [$-$] | 0.035 | 0.047 | 0.030 |
| $\partial$ R / $\partial$ B [$-$] | 0.101 | 0.013 | 0.226 |
| $\partial$ R / $\partial$ C [(m/dB)$^{-1}$] | 0.000 | 0.000 | 0.000 |

We use the Banner 2021 validation dataset to further illustrate the sensitivity of the S1 snow depth retrievals to the three parameters (Figure B1a-c). Changes in the $B$ and $C$ parameter have approximately linear effects on the change in mean scene-wide snow depth, with changes to $C$ impacting the snow depth retrieval the greatest. Changes in $A$ were generally linear until

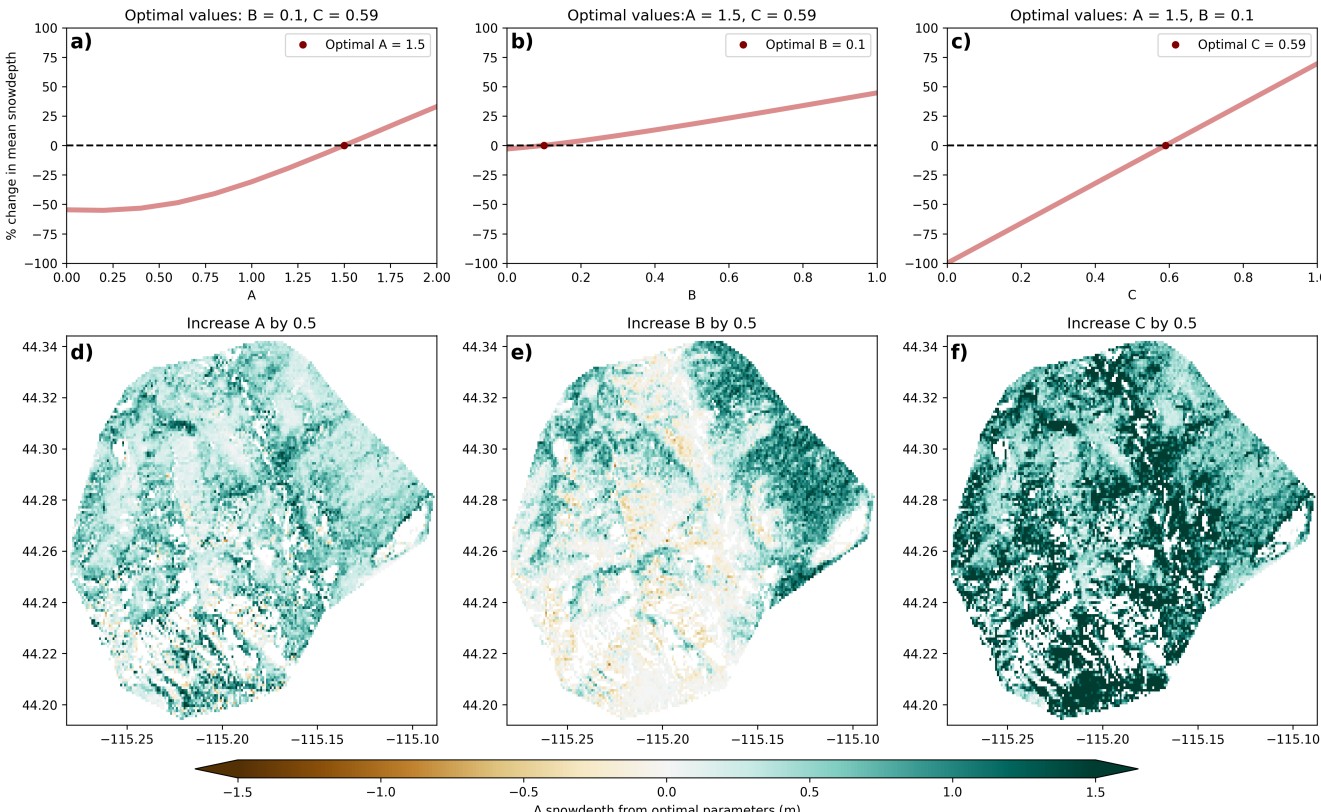

**Figure B1.** Percent change in scene-wide mean snow depth with varying A, B, and C parameters from optimized values (A = 1.5, B = 0.1, C = 0.59) for the Banner Summit 2021 site (a-c). Changes in S1 snow depth retrievals when increasing each parameter by 0.5 from the optimal value and keeping the other parameters at their optimal value (d-f).

≈0.5m where minimal snow depth changes were observed for further decreases in $A$. Increasing the $A$ parameter primarily impacts higher elevation areas with lower FC (Figure B1d), while increasing $B$ results in increased snow depths in lower elevation forested regions and actually causes slight a snow depth decrease in the less forested regions (Figure B1e). Modifying the scaling parameter $C$ affects all pixels, with the largest changes in regions with the greatest retrieved snow depths.

We found that the $C$ parameter has the greatest impact on RMSE (Table B1) and total retrieved snow depth (Figure B1),
indicating that $C$ is the most important parameter to optimize if minimizing scene-wide RMSE the primary consideration. Since $C$ simply scales values in the final step of the retrieval, this parameter can be optimized efficiently and should be adjusted first when applying this technique at a new site.

The $A$ and $B$ parameters had a much lower impact on scene-wide RMSE but controlled the spatial and temporal distribution of error. As such, practitioners optimizing these two parameters should evaluate the environmental characteristics of areas with
520 high RMSE. Optimizing $B$ may be most important in areas with greater forest cover, while conversely, optimizing $A$ may be more important in high-elevation areas with low forest cover. Importantly, A and B are not independent. Varying one will

cause the other to be mis-optimized, highlighting a potential weakness of this empirical model. A potential avenue to lower RMSE across a scene with varied environmental characteristics could be to apply two implementations of the algorithm, one optimized for areas with dense forest cover and another optimized for alpine areas with sparse vegetation.

While we did not evaluate the impact of outliers on parameter optimization, visual examination of 2021 S1 snow depth results at the Banner study site shows isolated areas of extreme snow depth along a rugged ridgeline at the center of the site (Figure 4b). These extreme outliers in snow depth likely caused a decrease in C parameter to and a corresponding decrease in snow depths in other areas, potentially introducing a negative bias in the S1 snow depth results. These outliers are also visible in (Figure 3b) with some outliers over 4+ meters apparent in the S1 retrievals but no in the lidar. To mitigate this issue, it may

be advantageous to perform parameter optimization on a high-confidence subset of the radar data, within elevation bands, or after outlier removal.

## B2    Wet snow parameters

The S1 algorithm has increased uncertainty over areas with wet snow (Lievens et al., 2022), which is why careful consideration must be taken to optimize the wet snow parameters to accurately classify wet snow. The three wet snow parameters described

in A (wet snow threshold, freeze snow threshold, and alternate wet snow flag) were not systematically optimized by Lievens et al. (2022). When attempting to optimize these parameters to minimize scene-wide RMSE, we found that no global optimum exists. Instead, we found that by increasing both the wet snow threshold and freeze snow threshold, RMSE decreases at the expense of a reduced number of retrieved snow depths as more pixels are masked out. This tradeoff is visualized in Figure B2.

During our analysis we found the original freeze snow threshold of +2dB to be overly conservative: pixels that we expect

to refreeze remain wet throughout the entire winter season, despite air temperatures dropping well below freezing. We noted a jump in backscatter in these pixels, but not enough to satisfy the +2dB threshold. Similar considerable (but not quite +2dB) jumps in VV backscatter during refreezing events were also observed by Lund et al. (2022). The +1dB freeze snow threshold we implemented resulted in a more realistic match with SNOTEL temperatures (Figure B3). Our selected parameters of a wet snow threshold of -3 dB, a freezing threshold of +2 dB, and choosing to keep the alternate wet snow flag enabled provide a

good compromise that results in an effective wet snow mask without overmasking to artificially boost algorithm performance.

With our optimized wet snow flagging parameters, the time series of wet snow and dry snow pixels matches well with the temperature and snow depth trends observed at the Banner Summit SNOTEL site (Figure B3). The spatial progression of melt agrees well with the SNOTEL temperature and snow depth measurements. Wet snow is observed in the early accumulation season (October through early December) when warmer daytime temperatures and mixed phase precipitation occur. Then, as

daytime temperatures progressively cool, water within the snowpack freezes and dry snow precipitation increases, expanding dry snow extent in the colder winter months (mid December through early March). Finally, warmer spring temperatures and increased shortwave radiation introduce surface melt in the snowpack, turning dry snow to wet snow beginning in mid-March until the snow melts away. This progression also coincides well with elevation: at Banner Creek, snow at lower elevations is more often observed as wet, and snow at higher elevations is more often observed as dry.

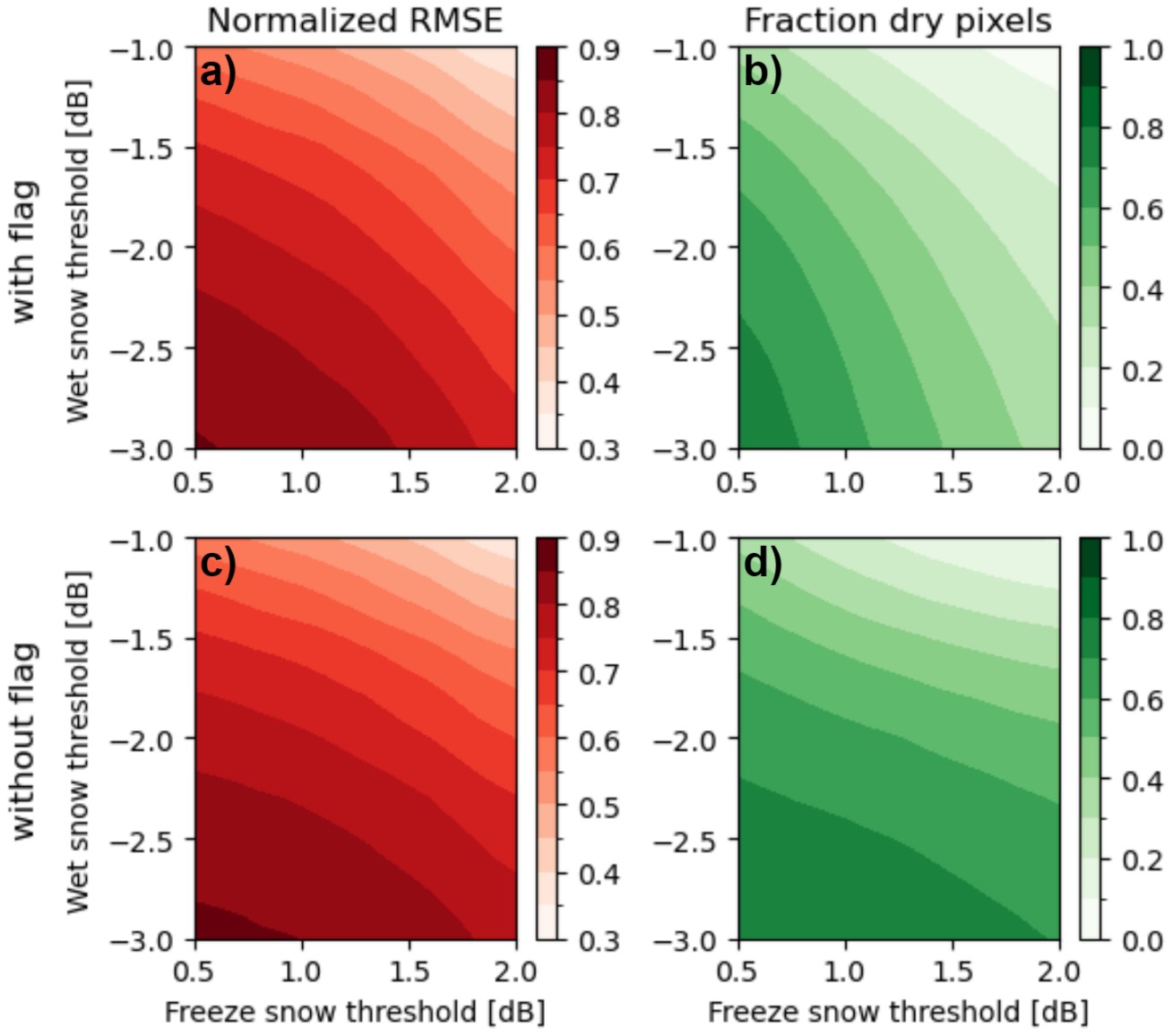

**Figure B2.** Binned mean normalized RMSEs (a & c) and fraction of dry pixels (b & d) for permutations of the newly wet and freezing thresholds and with (top row) and without (bottom row) the alternative wet snow flagging for the Banner 2021 lidar flight.

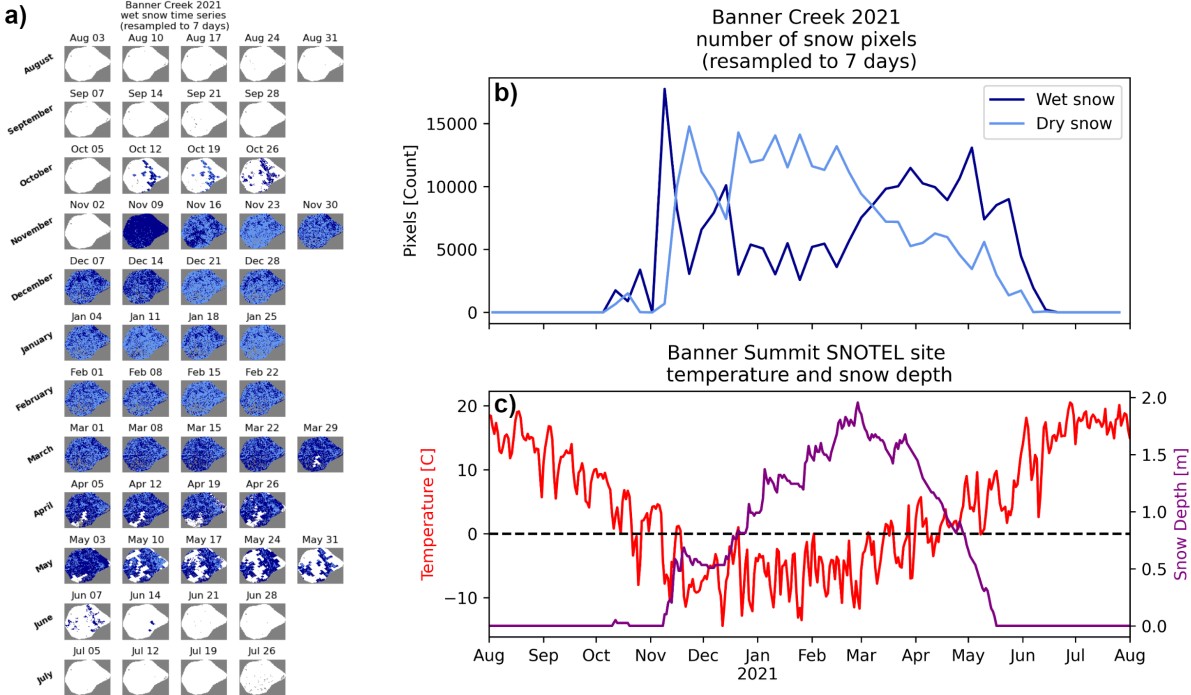

**Figure B3.** Time series of snow classifications for the 2020-2021 winter at the Banner study site (a) with no snow (white), dry snow (light blue), wet snow (dark blue). Number of dry vs. wet snow pixels in the scene from August 2020 to August 2021 (b), and temperature and snow depth values from the Banner Summit SNOTEL (c).

Though we make these recommendations for wet snow parameters, end users will have to make a final selection of parameter values that consider both local conditions as well as retrieval quality vs retrieval quantity. In this way, it is important to treat the wet snow parameter value selection just as carefully as the A, B, and C parameter value selection. Additionally, it is important to remember that this algorithm should only be used in the accumulation season (Tsang et al., 2022). Though the pixels flagged as wet snow can often follow reasonable snow depth trends, we suggest caution in the interpretation of these pixels, as changes
in snow depth are likely not due to changes in volume scattering as prescribed in the algorithm.

## Appendix C:  Sentinel-1 Imagery Used

All Sentinel-1 imagery used is presented below in Table C1

| Site | Satellite | Direction | Orbit | Acquisition Timing |
|---|---|---|---|---|
| Banner | S1B | ascending | 93 | 2019-08-05, 2019-08-17, 2019-08-29,2019-09-10 |
| | | | | 2019-09-22, 2020-01-20, 2020-02-01,2020-02-13 |
| | | | | Continued on next page |

| Site | Satellite | Direction | Orbit | Times |
|------|-----------|-----------|-------|-------|
| | | | | 2020-02-25, 2020-08-11, 2020-08-23,2020-09-04 |
| | | | | 2020-09-16, 2020-09-28, 2020-10-10,2020-10-22 |
| | | | | 2020-11-03, 2020-11-15, 2020-11-27,2020-12-09 |
| | | | | 2020-12-21, 2021-01-02, 2021-01-14,2021-01-26 |
| | | | | 2021-02-07, 2021-02-19, 2021-03-03,2021-03-15 |
| | | | | 2021-03-27 |
| Banner | S1B | descending | 71 | 2019-11-07, 2019-11-19, 2019-12-01,2019-12-13 |
| | | | | 2019-12-25, 2020-01-06, 2020-01-18,2020-01-30 |
| | | | | 2020-02-11, 2020-02-23, 2020-08-09,2020-08-21 |
| | | | | 2020-09-02, 2020-09-14, 2020-09-26,2020-10-08 |
| | | | | 2020-10-20, 2020-11-01, 2020-11-13,2020-11-25 |
| | | | | 2020-12-07, 2020-12-19, 2020-12-31,2021-01-12 |
| | | | | 2021-01-24, 2021-02-17, 2021-03-01,2021-03-13 |
| | | | | 2021-03-25 |
| Banner | S1A | descending | 71 | 2019-09-26, 2019-11-01, 2019-11-13,2019-11-25 |
| | | | | 2019-12-07, 2019-12-19, 2019-12-31,2020-01-12 |
| | | | | 2020-01-24, 2020-02-05, 2020-02-17,2020-08-03 |
| | | | | 2020-08-15, 2020-08-27, 2020-09-08,2020-09-20 |
| | | | | 2020-10-02, 2020-10-14, 2020-10-26,2020-11-07 |
| | | | | 2020-11-19, 2020-12-01, 2020-12-13,2020-12-25 |
| | | | | 2021-01-06, 2021-01-18, 2021-01-30,2021-02-11 |
| | | | | 2021-02-23, 2021-03-07,2021-03-19 |
| Banner | S1A | ascending | 93 | 2019-11-03, 2019-11-15, 2019-11-27,2019-12-09 |
| | | | | 2019-12-21, 2020-01-02, 2020-01-14,2020-01-26 |
| | | | | 2020-02-07, 2020-02-19, 2020-03-02,2020-08-05 |
| | | | | 2020-08-17, 2020-08-29, 2020-09-10,2020-09-22 |
| | | | | 2020-10-04, 2020-10-16, 2020-10-28,2020-11-09 |
| | | | | 2020-11-21, 2020-12-03, 2020-12-15,2021-01-08 |
| | | | | 2021-02-01, 2021-02-13, 2021-02-25,2021-03-09 |
| | | | | 2021-03-21 |
| Cameron | S1A | descending | 56 | 2020-08-02, 2020-08-14, 2020-08-26,2020-09-07 |
| | | | | 2020-09-19, 2020-10-01, 2020-10-13,2020-10-25 |
| | | | |  |

| Site | Satellite | Direction | Orbit | Times |
|------|-----------|-----------|-------|-------|
| | | | | 2020-11-06, 2020-11-18, 2020-11-30,2020-12-12<br>2020-12-24, 2021-01-05, 2021-01-17,2021-01-29<br>2021-02-10, 2021-02-22, 2021-03-06,2021-03-18<br>2021-03-30 |
| Cameron | S1B | ascending | 151 | 2020-08-03, 2020-08-15, 2020-08-27,2020-09-08<br>2020-09-20, 2020-10-02, 2020-10-14,2020-10-26<br>2020-11-07, 2020-11-19, 2020-12-01,2020-12-13<br>2020-12-25, 2021-01-06, 2021-01-18,2021-01-30<br>2021-02-11, 2021-02-23, 2021-03-07,2021-03-19<br>2021-03-31 |
| Dry Creek | S1A | descending | 144 | 2019-08-02, 2019-08-14, 2019-08-26,2019-09-07<br>2019-09-19, 2019-10-01, 2019-10-13,2019-10-25<br>2019-11-06, 2019-11-18, 2019-11-30,2019-12-12<br>2019-12-24, 2020-01-05, 2020-01-17,2020-01-29<br>2020-02-10,2020-02-22 |
| Dry Creek | S1A | ascending | 93 | 2019-11-03, 2019-11-15, 2019-11-27,2019-12-09<br>2019-12-21, 2020-01-02, 2020-01-14,2020-01-26<br>2020-02-07, 2020-02-19,2020-03-02 |
| Dry Creek | S1B | descending | 71 | 2019-11-07, 2019-11-19, 2019-12-01,2019-12-13<br>2019-12-25, 2020-01-06, 2020-01-18,2020-01-30<br>2020-02-11,2020-02-23 |
| Dry Creek | S1B | ascending | 166 | 2019-08-10, 2019-08-22, 2019-09-03,2019-09-15<br>2019-09-27, 2019-10-09, 2019-10-21,2019-11-02<br>2019-11-14, 2019-11-26, 2019-12-08,2019-12-20<br>2020-01-01, 2020-01-13, 2020-01-25,2020-02-06<br>2020-02-18,2020-03-01 |
| Fraser | S1A | descending | 56 | 2019-08-08, 2019-08-20, 2019-09-01,2019-09-13<br>2019-09-25, 2019-10-07, 2019-10-19,2019-10-31<br>2019-11-12, 2019-11-24, 2019-12-06,2019-12-18<br>2019-12-30, 2020-01-11, 2020-01-23,2020-02-04<br>2020-02-16, 2020-08-02, 2020-08-14,2020-08-26<br>2020-09-07, 2020-09-19, 2020-10-01,2020-10-13 |
| | | | |  |

| Site | Satellite | Direction | Orbit | Times |
|------|-----------|-----------|-------|-------|
| | | | | 2020-10-25, 2020-11-06, 2020-11-18,2020-11-30 |
| | | | | 2020-12-12, 2020-12-24, 2021-01-05,2021-01-17 |
| | | | | 2021-01-29, 2021-02-10, 2021-02-22,2021-03-06 |
| | | | | 2021-03-18,2021-03-30 |
| Fraser | S1B | ascending | 151 | 2019-08-09, 2019-08-21, 2019-09-02,2019-09-14 |
| | | | | 2019-09-26, 2019-10-08, 2019-10-20,2019-11-01 |
| | | | | 2019-11-13, 2019-11-25, 2019-12-07,2019-12-19 |
| | | | | 2019-12-31, 2020-01-12, 2020-01-24,2020-08-03 |
| | | | | 2020-08-15, 2020-08-27, 2020-09-08,2020-09-20 |
| | | | | 2020-10-02, 2020-10-14, 2020-10-26,2020-11-07 |
| | | | | 2020-11-19, 2020-12-01, 2020-12-13,2020-12-25 |
| | | | | 2021-01-06, 2021-01-18, 2021-01-30,2021-02-11 |
| | | | | 2021-02-23, 2021-03-07, 2021-03-19,2021-03-31 |
| LCC | S1B | ascending | 122 | 2020-08-01, 2020-08-13, 2020-08-25,2020-09-06 |
| | | | | 2020-09-18, 2020-09-30, 2020-10-12,2020-10-24 |
| | | | | 2020-11-05, 2020-11-17, 2020-11-29,2020-12-11 |
| | | | | 2020-12-23, 2021-01-16, 2021-01-28,2021-02-09 |
| | | | | 2021-02-21, 2021-03-05, 2021-03-17,2021-03-29 |
| LCC | S1A | descending | 100 | 2020-08-05, 2020-08-17, 2020-08-29,2020-09-10 |
| | | | | 2020-09-22, 2020-10-04, 2020-10-16,2020-10-28 |
| | | | | 2020-11-09, 2020-11-21, 2020-12-03,2020-12-15 |
| | | | | 2020-12-27, 2021-01-08, 2021-01-20,2021-02-01 |
| | | | | 2021-02-13, 2021-02-25, 2021-03-09,2021-03-21 |
| Mores | S1A | descending | 71 | 2019-08-09, 2019-08-21, 2019-09-02,2019-09-14 |
| | | | | 2019-09-26, 2019-10-08, 2019-10-20,2019-11-01 |
| | | | | 2019-11-13, 2019-11-25, 2019-12-07,2019-12-19 |
| | | | | 2019-12-31, 2020-01-12, 2020-01-24,2020-02-05 |
| | | | | 2020-02-17, 2020-08-03, 2020-08-15,2020-08-27 |
| | | | | 2020-09-08, 2020-09-20, 2020-10-02,2020-10-14 |
| | | | | 2020-10-26, 2020-11-07, 2020-11-19,2020-12-01 |
| | | | | 2020-12-13, 2020-12-25, 2021-01-06,2021-01-18 |
| | | | | 2021-01-30, 2021-02-11, 2021-02-23,2021-03-07 |

| Site | Satellite | Direction | Orbit | Times |
|------|-----------|-----------|-------|-------|
| | | | | 2021-03-19 |
| Mores | S1A | ascending | 93 | 2019-11-03, 2019-11-15, 2019-11-27,2019-12-09 |
| | | | | 2019-12-21, 2020-01-02, 2020-01-14,2020-01-26 |
| | | | | 2020-02-07, 2020-02-19, 2020-08-05,2020-08-17 |
| | | | | 2020-08-29, 2020-09-10, 2020-09-22,2020-10-04 |
| | | | | 2020-10-16, 2020-10-28, 2020-11-09,2020-11-21 |
| | | | | 2020-12-03, 2020-12-15, 2021-01-08,2021-02-01 |
| | | | | 2021-02-13, 2021-02-25, 2021-03-09,2021-03-21 |
| Mores | S1B | descending | 71 | 2019-11-07, 2019-11-19, 2019-12-01,2019-12-13 |
| | | | | 2019-12-25, 2020-01-06, 2020-01-18,2020-01-30 |
| | | | | 2020-02-11, 2020-08-09, 2020-08-21,2020-09-02 |
| | | | | 2020-09-14, 2020-09-26, 2020-10-08,2020-10-20 |
| | | | | 2020-11-01, 2020-11-13, 2020-11-25,2020-12-07 |
| | | | | 2020-12-19, 2020-12-31, 2021-01-12,2021-01-24 |
| | | | | 2021-02-17, 2021-03-01, 2021-03-13,2021-03-25 |
| Mores | S1B | ascending | 93 | 2019-08-05, 2019-08-17, 2019-08-29,2019-09-10 |
| | | | | 2019-09-22, 2020-01-20, 2020-02-01,2020-02-13 |
| | | | | 2020-08-11, 2020-08-23, 2020-09-04,2020-09-16 |
| | | | | 2020-09-28, 2020-10-10, 2020-10-22,2020-11-03 |
| | | | | 2020-11-15, 2020-11-27, 2020-12-09,2020-12-21 |
| | | | | 2021-01-02, 2021-01-14, 2021-01-26,2021-02-07 |
| | | | | 2021-02-19, 2021-03-03, 2021-03-15,2021-03-27 |

Table C1: Grouped Sentinel-1 observation dates by site, satellite, orbit, and direction

*Author contributions.* Conceptualization: ZH, HPM, GB, RP. Writing: ZH, RP, JT, GB, EG, DD, AM, HPM Analysis: ZH, RP, GB, DD, EG, AM, RB. Repository: ZH, RP, GB, EG. Funding Acquisition: HPM. Planning: ZH, HPM. Editing: ZH, HPM, GB, RP, JT, DD

*Competing interests.* The authors report they have no competing interests.

*Acknowledgements.* We wish to thank Dr. Tate Meehan for his suggestion of repository name. Dr. Joe Meyer for sharing SnowModel data for the East River Basin. Emma Marshall for helping with implementing the algorithm using Xarray. Dr. Randall Bonnell for useful discussions.

*Code availability.* The code for producing the S1 derived snow depths is available at: https://github.com/SnowEx/spicy-snow. Code for producing the figures and analysis is available at: https://github.com/ZachHoppinen/spicy-analysis.

*Financial support.* Lidar acquisitions and HM was supported by NASA grant 80NSSC21K0408. JT was supported by an appointment to the NASA Postdoctoral Program at the Goddard Space Flight Center (GSFC), administered by ORAU through a contract with NASA.

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
