# Peer review of "Evaluating Snow Depth Retrievals from Sentinel-1 Volume Scattering over NASA SnowEx Sites"

_EGUsphere, 2024_

## Referee Comment (RC3)

**Review of: Evaluating Snow Depth Retrievals from Sentinel-1 Volume Scattering over NASA SnowEx Sites**

**General Comments:**

This paper analyzes the performance of the Sentinel-1 snow depth retrieval developed by Lievens et al. on an independent dataset acquired during the SnowEx campaigns of 2019-20 and 2020-21.

This paper is well structured and well written. The fact that the authors have put in the work to make this algorithm open source is a big accomplishment and is of major relevance to the snow community. I really appreciate the amount of work this takes to publish such an open-source framework. This algorithm is highly debated in the snow community and was previously difficult to reproduce. This in itself warrants publication.

That said, given the empirical nature of this algorithm and the diverse set of geophysical properties of the SnowEx dataset, there are some major comments that I think should be addressed before publication in order to iron out the applicability of this algorithm.

**Major Comments:**

The first major comment relates to the interpretation of the cross-polarization signal with different snow properties. Unlike many previous publications, I appreciate that the authors try to define the scattering mechanisms that can depolarize the signal. The authors state themselves that the poor understanding of the radiative transfer mechanisms at C-Band is a weakness of this algorithm. This is very complex and needs deep understanding of the snow physical parameters but also of microwave radiative transfer physics of snow. I feel like some textbook information on signal depolarization within snow is needed in the introduction section (section 1.1). I also added comments which can help to improve the description (see minor comments).

I also think that the impact of snow on the S1 signal ratio is a bit simplified and could be improved by looking at the evolution of the polarizations individually. There seems to be more in Figure 8 than what is discussed, and looking at the polarizations individually could improve the interpretation of Figure 8.

This might be out of scope for this publication but one criticism that the Lievens et al. algorithm has is its empirical nature, and how a single set of parameters and one equation is used for a wide range of landscapes analyzed in this study. I would add a section to the Results section where different retrieved parameters for the different landscapes (forest cover, altitude, snow depth) would be analyzed. A simple comparison between the parameters presented here and the ones from Lievens, with regards to the landscape properties, would be very relevant. Some of this information is already included in the appendix and could be included in the results. The fact that there is imbalance in the data points for the different landscape properties indicates that some landscapes will show better results and others not.

Last major comment relates to some part of the results/discussion needs to the quantified rather than simply described qualitatively (see minor comments).

**Minor Comments:**

L.6: I would remove "we develop the first open-source software package" here and add the "first open-source software" part in the sentence of line 16. This sentence should focus on the algorithm implementation and its application. Moving this part will make the abstract more concise and avoid repeating information.

L.10 I would include the nRMSE here. 0.92m of RMSE when we don't know the mean snow depth is difficult to interpret. Alpine snowpacks can have several meters of snow.

L.12 Just a personal preference but I would call it the "cross- to co-pol backscatter ratio" (CCPR). The cross-pol ratio sounds more like VH/HV. But it is defined in the text so there is no major issue with it.

L.13 remove "cross".

L.14-15: I would reword the last part of the sentence. It is not clear to me is there is or not a correlation/relationship.

L.16: correct "frame work" to "framework".

L.24: I would change "the defining hydrologic variable of the seasonal snowpack" to something like: "an important hydrological variable of the seasonal snowpack". It's not the only important hydrological variable for water management.

L.33-34: I would break this sentence into two parts. I would put the correlation length part in the second sentence. In terms of correlation length, I imagine it relates to snow surface spatial auto-correlation. Correlation lengths is used in many ways in snow remote sensing, a bit more detail would be useful here.

L.36: I would specify here that it's for the NASA SWE product. The algorithm uses 19 and 37GHz where the 37 GHz saturates but the lower frequencies of AMSR-E and AMSR2 are sensitive to deeper snow than this.

L.52: I feel like the paragraph with the overview of SAR methods should site the review paper of Tsang et al. (2022) which is cited further in the paper.

L.53: change "a type of" to "an".

L.57: add "and polarization" to "SAR signal's frequency".

L.57: replace "for retrievals of" to "to retrieve".

L.67: delete "a".

L.72: no need to specify the frequencies here but it can span further than 40 GHz

L.74: Needs a textbook reference here.

L.75: The citations are in a sentence alone.

L.79: delete substantial.

L.79: replace "the depth of microwave penetration" to "microwave penetration depth".

L.96: there is also second order scattering, i.e. the impacts of multiple scatterers from multiple interfaces in different orientations (roughness) which increases the cross-polarization backscatter. This effect could be highlighted by a thicker snowpack.

L.97: these anisotropic clusters are rarely at the surface and are created within the snowpack via metamorphism. It requires a strong temperature gradient which can be amplified by a thicker snowpack. A good reference for the impacts of the anisotropic snow grains is the publication of Picard et al. (2022) on "microwave snow grain size".

Figure 1: Not exactly how a SAR system works, here it seems like the emitter and receiver are on two different platforms. Also, the volume scattering at C-Band is a very small portion of the backscattered signal. The other components should be highlighted.

L.103: Again, the cross- to co-pol ratio is more intuitive to me.

L.122: IW could in theory be HH+HV but VV+VH is the preferred polarizations over land. I would rephrase: [...] IW swath mode, dual-polarized vertical transmit, and vertical/horizontal receive (VV+VH).

L.130: multi-looking is used in many different flavors for SAR applications. Do you mean a 3x3 block average, which reduces the 30m resolution to 90 m resolution?

Figure 2: The boxes are small. Subplots of close ups of the three bigger AOIs would be more useful here, i.e. three maps and their bounding boxes could be shown in the broader map at the top right.

L.163-164: These lines seem out of place. Delete?

L.180-181: The coefficient values are results and could be discussed with regards to the Lievens parameters. I would move this to the results section in a new section on the parameterization. Some info from the appendix could also be included in this new section.

L.186-195: This section should be in results/discussion

Table 2: I would include the nRMSE as well.

L.233-234: Given the empirical nature of the retrieval method, it would be useful to determine what is the fraction of pixels that belong to the different "classes". This alone could explain these results if you have an imbalance dataset where most pixels/measurements have deeper snow, moderate FC, and are at higher elevation.

Not surprised with the coarser resolution since you remove the spatial variability of snow and landscape properties.

L.241-243: This needs to be quantified and not just analyzed qualitatively.

L.246-248: Also needs to be quantified.

Figure 8: This figure shows more than what is discussed here... The fact that <0.5 m you have no change in deltaCR indicates that you have no significant change to the signal coming from the snow. Then the drop seems to indicate that there is attenuation of the deltaCR with .5 to 1m of snow. Finally, with 1m of snow and more, there seems to be an added contribution to the deltaCR (volume scattering?). That figure in itself is not sufficient to explain the different scattering mechanism in action. The same figure but for VH and VV separately would help identify which mechanism dominates and its link to snow depth.

L.262: It's the first time SNR is mentioned. It's an important aspect of SAR retrieval algorithm. I imagine you mean an increase in SNR for VH (VV should be fairly stable in dry snow conditions).

L.263-266: No-correlation needs to be quantified.

L.267: I would even reduce it to snow-air and snow-soil interfaces since the dielectric contrast between two snow layers is very low, and vertical polarization is less sensitive to interface reflectivity than horizontal polarization.

L.270-271: This should be included in the introduction with the SAR sensitivity to snow.

L.277-278: This could be due to the low number of sampling points with low FC to retrieve the new empirical parameters of Eq. 1.

L.282: SNR could be a way to mask out some pixels. Wet snow will usually show a very low VH signal which, depending on the surface roughness, would be close to the noise floor.

L.307-308: I was under the assumption that the parameters were fixed using the 90m scale data. This second point seems to indicate that the parameters were changed for different resolutions. Please clarify.

L.322: Not necessarily true, this is true for passive microwave signals. Madore et al. (2023, see chapter 6) have shown that K-Band signal could be sensitive to several meters of snow which is at a higher frequency than Ku-Band.

L.331: Retrieving eq. 1 parameters on dry snow conditions tested? or all data was used.

L.336: but there seems to be a detectable decrease between <.5 and .5-1 m of snow. Any possible explanation?

L.343: That is a major weakness of this algorithm. It could be included in the introduction since it was mentioned by most reviewers publicly in the original Lievens paper in The Cryosphere.

L 350: Including passive microwaves is very difficult in high topography areas with its coarse resolution.

L.350: I would also include polarimetric SAR approaches.

L.353-354: I agree that this is more important than trying to tune an empirical algorithm for the infinite number of landscapes we can find globally.

---

## Referee Comment (RC4)

General

Lievens et al. (2019)'s work showed very good results for estimating snow, and had some debate in the snow community on why/how it works. This works is very valuable to verify the retrieval finding.

Major comments

The fact that they showed the results don't match well enough the LIDAR data is very important. However, the fact that the Hoppinen's snow depth results also differ from Lievens, is not a good sign and need to find the reason and maybe find A, B, C, they have used. On the other hand, I would like to see the same verification that Lievens did, i.e. comparing with in situ data. I think the authors can compare the estimated snow depth with WUS snow depth and see if they get the same results as Lievens or not. If they did not then it is safe that the results are not verified. If they do, it may be errors in LIDAR data in mountainous regions or something.

Here are some specific comments

Line 187: there should be more investigation of this much low correlation. I suspect even changing A, B, C parameters will change thing much.

Line 241/Figure 7: I think it is not a correct comparison, you need to use the S1 CR at in situ locations for comparing with measured snow depth

Figure 5: it is very misleading. The histograms should have the same normalized values. For instance, 5a blue and orange has almost the same maximum but blue is very narrow. I assume compareing mean and std of will give a better and more quantitative comparison. So. I suggest to generate the same plot for mean and std for different x-axis parameter. This way you can show the results for all sites in the same figure too.

Minor comments:

Line 61: "radar approaches are more directly related to SWE than depth" : the only radar approach that is directly related to SWE is InSAR whereas two frequency amplitude ones are related to snow depth. Need to correct this sentence.

Line 123: 2-6 days revisit for Sentinel-1 is too much. I guess it assumes both Sentinel-1 are on and making observation. We know that this is not the case everywhere and all the time. Need to fix this. Also it needs to be clarified if it used both ascending descending observations or just one direction.

Line 136: I am not sure what you did here. I assume you want to compensate the effect of incidence angle for overlaps. Could you please clarify what you exactly did here and how it is supposed to help you.

Line 189: remove extra data

Figure 2b x-axis should be snow depth, remove lidar

Figure 2b: it is not clear what dashed lines inside the histograms show.

Line 263: higher volume scattering and higher SNR: the SNR is not defined, and we suggest using another term, as SNR normally refers to radar received signal compared to received noise. I think you are using S and N with a different definition. If so, please use other term. Also, more volume scattering doesn't necessarily mean more depolarization, for instance for an isotropic volume it doesn't make it depolarized.

Line 284: need to provide a reference for east (more wind-deposit snow) and west (more direct solar radiation) facing comments.

Line 305: you need reference for this. I don't think orbital error/variation in ground/vegetation properties affect the "noise" (not snow backscattered power). Normally speckle noise is the part that gets improved by taking looks.

Line 352: remove extra be

Line 365 a should be an

---

## Author Comment (AC1)

June 24th, 2024
Evaluating Snow Depth Retrievals from Sentinel-1 Volume Scattering over NASA SnowEx Sites
By: Z. Hoppinen et al.

Reviewer comments are shown in black. Responses are in blue.

**Response to Reviewer #2**

**General Comments:**

This article reports on the evaluation of an algorithm for retrieval of snow depth based on C-band backscatter intensity images of the Sentinel-1 mission. The first version of the retrieval algorithm, presented by Lievens et al. (2019), applied an empirical change detection method using temporal changes of the cross- to co- polarization backscatter ratio (VH/VV) to compute a snow index that is rescaled by means of empirical parameters in order to obtain the snow depth. A modified version of the snow depth algorithm was applied for snow depth retrievals over the European Alps (Lievens et al., 2022). This version of the algorithm employs the VH/VV ratio and the VV backscatter intensity, accounts for the fractional forest cover, and uses several empirical scaling and weighting parameters. A critical constituent of the algorithm is the determination of empirical scaling factors that relate the backscatter intensity to snow depth.

A comprehensive independent validation of the algorithm has been lacking by now. The work by Hoppinen at al. addresses this open issue, evaluating the performance of Sentinel-1 snow depth maps derived by means of the 2022 version of the algorithm. Reference data for performance assessment are available from nine high-resolution airborne lidar acquisitions over extended study sites in the Western US, an excellent data set for algorithm validation. The scaling factors are derived from a subset of these data by optimizing the correlation coefficient and minimizing the mean absolute error.

The paper is a valuable contribution to the topic of SAR-based approaches for mapping snow depth and snow mass. It addresses a very relevant question in this context. The data analysis, results and conclusions are well described and conclusive. However, there are still some issues to be checked and clarified, addressed below. In particular Section 1.1 on the theoretical background needs major revision. Besides, information on the available Sentinel-1 coverage (orbits, repeat coverage) would be of interest.

We thank you the reviewer for these in-depth comments. We believe the manuscript changes, highlighted below, address all concerns and result in a much improved manuscript.

**Specific comments**

1. Section 1.1 SAR volume scattering snow depth retrieval theory:

   This section refers to the radar signal interaction with snow. The description of processes having an impact on the C-band backscatter signal for snow over ground is incomplete and lacks quantitative information. Volume scattering cannot be considered as a stand-alone process for deriving physical snow parameters from backscatter intensity. In support of interpretation and discussion of the results of the study, I recommend including a concise description of the main contributions to the observed backscatter signal (including not only snow, but also ground and vegetation) and their impact on retrievals of snow depth. Because large parts of the study sites are covered by forest, the impact of forests on C-band signal propagation should be addressed. Figure 1 needs to be revised. The information to be conveyed by this figure is unclear. It implies that the incoming radar signal is reflected within the snow volume as a main source and the signal increases directly with increasing snow depth. Besides, the figure shows incoming and reflected beams in bistatic configuration.

   We have made major revisions to our initial discussion of SAR theory, including specifically mentioning the various other contributors to SAR backscatter over snow-covered ground (line 84-86). We present a generalized discussion of which scatterers are relevant and the relative importance of these scattering factors at non-grazing incidence angles (line 88-93). We have also highlighted that C-band SAR snow depth estimation methods rely on the assumption that there are minimal changes in vegetation or ground surface scattering characteristics throughout the snow covered period of observation (line 100).

   We have also modified Figure 1 to include non-snow related contributors to volume scattering and signal depolarization and have separated the two acquisitions to clarify that they are two monostatic time acquisitions.

2. Line 12 (Abstract): the term "cross-polarization backscatter ratio" should be defined at its first usage.

   Added formula after this usage to define.

3. Line 13 (Abstract): Please provide a number for "significant correlation"

   This is quantified by wilcoxon-box tests with a p-value below 0.0001. We have rephrased this sentence due to other reviewer comments (Reviewer 3, minor comment #5) that it was unclear and removed the "significant correlation" as part of those changes.

4. Line 99, 100: "as new snow increases the cross-polarized energy that is backscattered toward the sensor" This is not in accordance with experimental data and theory. Due to the low C-band scattering albedo of fresh snow the backscatter signal of a medium below with higher scattering albedo (e.g. coarse-grained metamorphic snow, refrozen snow) is attenuated when propagating through fresh snow.

Agreed, the choice of "new snow" here was misleading. We have changed "new snow" to "snow depth increases" to better capture that the current theory and literature is unclear on the exact rationale for the increased depolarization by snow at C-band, but that new snow is certainly not the only factor.

5. Line 125: Please check these two references: "Frerebeau et al., 2023; Lebrun et al., 2020"; both refer to "Dose Rate Estimation from in-Situ Gamma-Ray Spectrometry" and not to SAR.

Changed to correct citations for GAMMA and Sentinel-1 for GAMMA

6. Line 126: "European Space Agency, 2021" missing in the reference list
It was incorrectly abbreviated to Agency, E.S. We have changed this to "European Space Agency" in references.

7. Line 130: The speckle-related uncertainty of the selected grid size would be of interest.

We have chosen our grid size(s) to match Lievens et al. (2022). Future work on SAR speckle uncertainty as a function of grid size is an interesting topic, but we believe this to be outside the scope of this analysis and would distract from the primary conclusions. We have added a sentence to future work discussing the need for this analysis.

8. Line 133 to 135: The backscatter intensity at different incidence angles is not an "artifact" but contains relevant information related to physical properties of a medium which is suppressed of data with different incidence angles are merged.

We have changed "artifact" to "differences" and added a future work sentence suggesting the need to assess the relationship between incidence angle and backscatter/depolarization (line 411).

Since we are following the methods in Lievens et al. (2019, 2022) which use this incidence angle normalization we also performed this normalization to replicate the published method.

9. Line 175 to 177: According to this information a single S1 image per lidar acquisition data set (a subset of the total S1 data set) is used for deriving optimized scaling parameters. However, the snow depth retrieval algorithm is not based on single images but on changes of backscatter intensity in time over an extended period (Appendix A).

   The single S1 image is the result of the cumulative time series to that point and should be capturing information from all the relevant S1 backscatter changes that had snow coverage in the IMS data.

10. Line 208, 209: From the histograms in Fig. 3b, the agreement between the medians of these three sites is not obvious (due to the log-scaling). Besides, Banner 2020 and Fraser 2020 show a rather high negative bias for average snow depth (Table 2).

    We agree this line is over-optimistic and qualitative. Changed to "At most other sites snow depth is strongly underestimated by the S1 retrieval and retrieved snow depths exhibits much larger dynamic ranges compared to lidar" (line 234-235).

11. Line 209: The acronyms "ICC", and in Fig 3b "LCC", refer probably to Little Cottonwood?

    Corrected.

12. Fig. 3a: The linear trendline (on which the correlation coefficient is based) should be included, as its slope is an indication for the S1 sensitivity in respect to snow depth.

    Added in linear trend line along with equation.

13. Line 240 to 244 and Fig.7: The SNOTEL snow depth times series should be compared to the Sentinel-1 CR data of areas in the vicinity of the SNOTEL stations rather than to the site-wide mean cross ratio. Surface elevation has a major impact on the state of the snowpack and its backscatter properties.

    Changed figure 7 to include a 1 km buffer around the SNOTEL to get CR. Changed appropriate lines of text and caption to reflect this change.

14. Line 245 to 248 and Fig. 8: The two classes with deep snow (2.5-3, 3+) showing a distinct rise in delta-CR comprise only 1.2 % of the total sample. Hardly a suitable basis for a statistically significant conclusions regarding the retrieval performance for deep snow.

While these two classes are a small portion of the pixels, they also represent ~800 pixels and we believe that plotting them provides useful information about the relationship between CR and snow depth. We also quantified this relationship in figure 8 and found a significant change in means (line 272-275). Future work exploring a dataset with more deep snow lidar acquisitions would be valuable.

15. Page 14, Fig. 4 caption: the labels for FC and elevation in the figures and caption are mixed up.

    Fixed.

16. Page 15, Fig. 5: These histograms are probably also log-scaled.

    These are linear not log-scaled.

17. Line 252: The correlation coefficient is not a suitable parameter for assessing the performance of the spatial distribution.

    We chose this statistic to allow for direct intercomparison with Lievens et al. (2022).

18. Line 255: "Frasier" typo

    Fixed.

19. Line 261-262: Fig. 6a shows a decrease of the relative error with lidar snow depth, but the absolute error increases with snow depth for snow depth > 1 m. Also, in Fig. 5a the S1 and lidar histogram for snow depth > 2 m show the largest disparity. This is not a clear evidence for improved performance for deep snow.

    It is certainly suggestive when combined with Figure 7 and 8, previous work showing more response at deeper snow depths, and a theoretical understanding of C-band volume scattering. Clarified that this understanding relies on not only on Figure 6a but other previous work and figure 7 and 8 in line 384.

20. Line 267: "SAR signals primarily interact with layers within the snowpack rather than individual snow grains". Please check this statement. The scattering elements within layers are grains and grain clusters.

    Added "anisotropic grains and grain clusters". The cited Brangens et al. paper suggests that scattering at layer interfaces within the snowpack also play a role.

21. Line 267-271: Experimental data on propagation losses show for C-band power penetration length in dry seasonal snow typical numbers in excess of 10 m. Consequently, backscatter contributions of the subnivean ground and snow/ground interaction play also a role.

Agreed though skin depths for C-band into soil are relatively limited. Added in this contribution to the radar theory section (line 88) and into Figure 1.

22. Line 304-305: The Sentinel-1 orbit accuracy is very high so that orbit errors do not play any role. For estimating the impact of SAR speckle, estimates for the speckle-related uncertainty would be useful.

Removed orbital errors from this sentence. See response to #7 for speckle.

23. Line 317-318: " ... S1 snow retrievals agree best with lidar snow depth measurements in regions with snow packs deeper than 1.5 m …" this refers to the local snow depth relative to the mean value, not to the magnitude of snow depth. This should be stated. See comment line 261.

Not quite sure what you mean "local snow depth relative to the mean value". Do you mean we show relative error rather than absolute error?

24. Line 321-323: Ku-band and X-band are mentioned here. Why not L-band, for which the InSAR phase delay is applicable for SWE retrievals also in deep snow?

Since this paper has primarily focused on backscatter based approaches for which L-band is unsuitable we omitted mentioning it here. See revised introduction paragraph (lines 62-73) where we have added a new paragraph discussing other SAR retrieval techniques including many time-of-flight based approaches for which L-band is suitable.

25. Line 324 ff, Section 4.2: The analysis of the CR time series suffers from the spatial disparity between point measurements (SNOTEL) and spatial average S1 data of large test sites extending over different elevation zones (see comment line 240ff).

See response to comment line 240. We have changed the line in the figure/analysis to show the CR behavior in a 1km box around the SNOTEL sites instead of the mean-site CR and altered the appropriate text.

26. Line 373 to 376: A conclusive time series analysis of CR is lacking. See the comment above.

See response to comment line 240.

27. Line 375: Please provide numbers for "significant relationship".

This relationship comes from notched boxplots so we can say the increase has a p-value < 0.05. We have further quantified with a wilcoxon signed rank test (line 273-275)

Appendix B:

28. Line 459, Table B1: Units?

Added units to this table and text. Omitted units on text that are completely dimensionless.

29. Line 466, 467: "Since C simply scales values in the final step of the retrieval, this parameter can be optimized efficiently and should be adjusted first when applying this technique at a new site". This indicates that for any site an individual calibration of the scaling parameters is needed to obtain useful results. To this end representative and reliable snow depth data from other sources are needed. Furthermore, due to interannual changes in permittivity and structural properties of the snow cover ground, the value of the scaling parameter may change from year to year. This questions the feasibility of the retrieval approach for regular applications over extended areas.

Correct and one of the weaknesses of this algorithm. We chose to move this to the appendix since we advocate for an entirely new algorithm but originally this was a major weakness we planned to highlight.

30. Line 484, Fig. B1: Please check the sign for the % change in snow depth in Figs a, b, c. Would not C = 0.0 result in zero snow depth (-100%), rather than in +100% ?

Thank you for catching this error. We had the signs reversed for the three subpanels in this figure. This has been corrected in the revised version and C = 0.0 does result in  -100% snow depth as expected.

31. Page 28, Fig.B2: Figs. B2a and B2c show low RMSE for the wet snow threshold -1 dB and high RMSE for the threshold -3dB. As a lower wet snow threshold reduces the number of misclassifications for dry snow, the opposite behaviour is expected.

Respectfully, we believe the expected behavior is shown: the aggressive wet snow threshold of -1 dB will mask out more pixels, misclassifying a greater amount of truly dry pixels as instead falsely wet and thus removing all but the driest pixels from the RMSE analysis. The more conservative -3 dB will misclassify more truly wet pixels as falsely dry, including them in the RMSE analysis. With this more conservative -3 dB threshold, the misclassification of a greater number of truly wet pixels as falsely dry will drive the RMSE up, as we expect reduced algorithm performance in wet snow conditions.

---

## Author Comment (AC2)

June 25th, 2024
Evaluating Snow Depth Retrievals from Sentinel-1 Volume Scattering over NASA SnowEx Sites
By: Z. Hoppinen et al.

Reviewer comments are shown in black. Responses are in blue.

**Response to Reviewer #4:**

**General Comments:**

Lievens et al. (2019)'s work showed very good results for estimating snow, and had some debate in the snow community on why/how it works. This works is very valuable to verify the retrieval finding.

Thank you for these valuable comments. Please see our responses below.

**General Comments:**

The fact that they showed the results don't match well enough the LIDAR data is very important. However, the fact that the Hoppinen's snow depth results also differ from Lievens, is not a good sign and need to find the reason and maybe find A, B, C, they have used. On the other hand, I would like to see the same verification that Lievens did, i.e. comparing with in situ data. I think the authors can compare the estimated snow depth with WUS snow depth and see if they get the same results as Lievens or not. If they did not then it is safe that the results are not verified. If they do, it may be errors in LIDAR data in mountainous regions or something.

The main challenge with comparing the two snow depth datasets is that the published methods in Lievens et al. (2022), which we follow here as closely as possible from their description in the paper, differ from the actual methods used to generate the data and figures in the paper as well as the data available in the C-SNOW repository (https://ees.kuleuven.be/eng/apps/project-c-snow-data/). This discrepancy was confirmed to us by personal communication with Dr. Hans Lievens which we cite in our manuscript (line 216). Additionally there is little to no documentation or metadata available on C-SNOW to clarify which data version relates most closely to the published 2022 methods.

Lievens et al. (2022) report A, B, and C parameter values optimized on a dataset from the European Alps. We initially used the same values in our calculations but unexpectedly large errors prompted us to re-optimize the algorithm for the different snow climate in the western US. We followed the optimization methods described in Lievens et al. (2022) and included relevant details in the main text (lines 203-205) and Appendix B. We found that the algorithm performed

better using our WUS-optimized parameters than the original set of Alps-optimized parameters. We also do compare our re-optimized algorithm to in situ SNOTEL data (Figure 7), although the in situ network is less dense across our validation sites than what is available in the Alps. Lidar snow depth uncertainties are on the order of 2-20 centimeters (Table 1) and cannot fully explain the ~meter scale errors we find in this study.

While the differences between our results and the C-SNOW data are concerning, we have recreated the published methods from Lievens et al. (2019, 2022) and have established an open platform for future users to iterate and evaluate this C-band depolarization technique. We hope that going forward the original authors can release the full details necessary to exactly replicate the C-SNOW data. If the missing information and full algorithm were made available we would gladly implement them into our open-source version.

1.  Line 187: there should be more investigation of this much low correlation. I suspect even changing A, B, C parameters will change thing much.

    See response above as well as a discussion of parameter optimization in Appendix B.

2.  Line 241/Figure 7: I think it is not a correct comparison, you need to use the S1 CR at in situ locations for comparing with measured snow depth

    Agreed. We have changed this analysis to use S1 CR for a 1km buffer around the SNOTEL stations (Figure 7) and made appropriate changes to the caption and text (lines 266).

3.  Figure 5: it is very misleading. The histograms should have the same normalized values. For instance, 5a blue and orange has almost the same maximum but blue is very narrow. I assume compareing mean and std of will give a better and more quantitative comparison. So. I suggest to generate the same plot for mean and std for different x-axis parameter. This way you can show the results for all sites in the same figure too.

    We disagree that this figure is misleading. In panel 5a the distributions are generated by grouping the lidar snow depths themselves. So we impose strict bounds on the lidar (blue) snow depths and compare the corresponding S1 (orange) snow depths, which have a much wider range of values and therefore wider/shorter distribution curves. When the selected lidar and S1 distributions have similar ranges (e.g. 75-100% forest cover in panel 5b or 500m spatial resolution in panel 5f) the relative height of the distributions is closer. The 25th, 50th, and 75th percentiles are notated with dashed lines in the distributions for a quantitative comparison; we have added this information explicitly to the figure caption.

**Minor comments**

4. Line 61: "radar approaches are more directly related to SWE than depth" : the only radar approach that is directly related to SWE is InSAR whereas two frequency amplitude ones are related to snow depth. Need to correct this sentence.

Agreed. We have made significant revisions to this section and believe we have corrected this poor/incorrect wording. Specifically see lines 62-71.

5. Line 123: 2-6 days revisit for Sentinel-1 is too much. I guess it assumes both Sentinel-1 are on and making observation. We know that this is not the case everywhere and all the time. Need to fix this. Also it needs to be clarified if it used both ascending descending observations or just one direction.

Revised this sentence to clarify we are talking about the spatial overlap of S1 swaths from different orbit geometries imaging a point every 2-6 days, while the revisit interval for a matching orbit geometry is either 6, 12, or 18 days. (line 148-150).

Also clarified we used all available images. Appendix A contains details on the separation and normalization of images clarifying that we used ascending, descending, S1A, and S1B.

6. Line 136: I am not sure what you did here. I assume you want to compensate the effect of incidence angle for overlaps. Could you please clarify what you exactly did here and how it is supposed to help you.

We are following the methods described in Lievens et al. (2019, 2022). There is a full description of this choice in Lievens et al. (2019) and in Appendix A of our manuscript.

7. Line 189: remove extra dataFigure 2b x-axis should be snow depth, remove lidar

Removed extra "data" and adjusted x-axis label.

8. Figure 2b: it is not clear what dashed lines inside the histograms show.

We assume this comment is in reference to Figure 3b. Added to caption to clarify these refer to 25th, 50th, and 75th percentiles.

9. Line 263: higher volume scattering and higher SNR: the SNR is not defined, and we suggest using another term, as SNR normally refers to radar received signal compared to received noise. I think you are using S and N with a different definition. If so, please use

other term. Also, more volume scattering doesn't necessarily mean more depolarization, for instance for an isotropic volume it doesn't make it depolarized.

We have changed to "volume scattering" to "depolarization" (line 294). Added definition of SNR term as we are using it here (lines 250).

10. Line 284: need to provide a reference for east (more wind-deposit snow) and west (more direct solar radiation) facing comments.

Edited to clarify that south slopes receive more solar radiation and west slopes receive more solar radiation during the warmer afternoon. Added citation for east slopes (more wind-deposited snow) for the western US.

11. Line 305: you need reference for this. I don't think orbital error/variation in ground/vegetation properties affect the "noise" (not snow backscattered power). Normally speckle noise is the part that gets improved by taking looks.

Removed the reference to orbital errors here.

12. Line 352: remove extra be

Done

13. Line 365 a should be an

Done

---

## Author Comment (AC3)

June 25th, 2024
Evaluating Snow Depth Retrievals from Sentinel-1 Volume Scattering over NASA SnowEx Sites
By: Z. Hoppinen et al.

Reviewer comments are shown in black. Responses are in blue.

**Response to Reviewer #3: Benoit Montpetit**

**General Comments:**

This paper analyzes the performance of the Sentinel-1 snow depth retrieval developed by
Lievens et al. on an independent dataset acquired during the SnowEx campaigns of 2019-20 and
2020-21.

This paper is well structured and well written. The fact that the authors have put in the work to
make this algorithm open source is a big accomplishment and is of major relevance to the snow
community. I really appreciate the amount of work this takes to publish such an open-source
framework. This algorithm is highly debated in the snow community and was previously difficult
to reproduce. This in itself warrants publication.

That said, given the empirical nature of this algorithm and the diverse set of geophysical
properties of the SnowEx dataset, there are some major comments that I think should be
addressed before publication in order to iron out the applicability of this algorithm.

We thank the reviewer for their insightful comments. The comments were thorough and helpful
and have certainly improved the writing and analysis. The suggestion to include delta VV and
VH in the boxplots was especially well worth the work.We have highlighted changes made to the
manuscript below in response to the reviewer's suggestions.

**Major Comments:**
The first major comment relates to the interpretation of the cross-polarization signal with
different snow properties. Unlike many previous publications, I appreciate that the authors try to
define the scattering mechanisms that can depolarize the signal. The authors state themselves that
the poor understanding of the radiative transfer mechanisms at C-Band is a weakness of this
algorithm. This is very complex and needs deep understanding of the snow physical parameters
but also of microwave radiative transfer physics of snow. I feel like some textbook information
on signal depolarization within snow is needed in the introduction section (section 1.1). I also
added comments which can help to improve the description (see minor comments).
      We appreciate these comments and strongly agree that much more research is necessary
to understand the relationship between different snow properties and signal depolarization. As

we felt the primary focus of this paper was to evaluate a specific algorithm and platform for snow depth retrievals we didn't want to get too deeply into the DMRT and theory behind C-band depolarization but instead to give a general understanding of the theory and some citations for those interested in the radiative theory. We have added in more discussion of this depolarization theory and citations for some of the fundamental works on this depolarization theory (line 88, 93, 97) along with a few textbook and review papers (line 71, 88).

I also think that the impact of snow on the S1 signal ratio is a bit simplified and could be improved by looking at the evolution of the polarizations individually. There seems to be more in Figure 8 than what is discussed, and looking at the polarizations individually could improve the interpretation of Figure 8.

We have made significant revisions to Figure 8 in response to your minor comment #34 and agree that an analysis of the polarization individually for this part was a very valuable analysis and have consequently also revised our discussion of this Figure (line 377-384).

This might be out of scope for this publication but one criticism that the Lievens et al. algorithm has is its empirical nature, and how a single set of parameters and one equation is used for a wide range of landscapes analyzed in this study. I would add a section to the Results section where different retrieved parameters for the different landscapes (forest cover, altitude, snow depth) would be analyzed. A simple comparison between the parameters presented here and the ones from Lievens, with regards to the landscape properties, would be very relevant. Some of this information is already included in the appendix and could be included in the results. The fact that there is imbalance in the data points for the different landscape properties indicates that some landscapes will show better results and others not.

We agree that the empirical nature of this algorithm is a challenge for future global implementation and specifically address the need for either a more complex model that can better capture these landscape parameters or is more physically based (lines 397-404). We originally had the analysis of parameters from Appendix B in the main body of the paper but decided to move the major of that discussion to the Appendix since it confused the primary takeaways of the paper and are of interest to a much smaller group of researchers than the primary evaluation of this S1 algorithm. Overall, we believe that the algorithm needs such dramatic changes that a specific analysis of the A, B, and C parameters specifically is probably not a helpful comparison for most researchers.

Last major comment relates to some part of the results/discussion needs to the quantified rather than simply described qualitatively (see minor comments).

We have quantified these relationships. See our responses to minor comments #32, 33.

**Minor comments**

1. L.6: I would remove "we develop the first open-source software package" here and add the "first open-source software" part in the sentence of line 16. This sentence should focus on the algorithm implementation and its application. Moving this part will make the abstract more concise and avoid repeating information.

   We believe these are two separate ideas. One is an open source algorithm that the whole community can work with and evaluate openly. The other is an open-source framework for future work on C-band volume scattering.

2. L.10 I would include the nRMSE here. 0.92m of RMSE when we don't know the mean snow depth is difficult to interpret. Alpine snowpacks can have several meters of snow.

   We chose to revise this wording to highlight that this RMSE is significantly worse than the requirements for remotely sensed observations of snow which are usually given as an RMSE (Line 10).

3. L.12 Just a personal preference but I would call it the "cross- to co-pol backscatter ratio" (CCPR). The cross-pol ratio sounds more like VH/HV. But it is defined in the text so there is no major issue with it.

   We find the phrase cross- to co-pol backscatter ratio to be a pretty clear and succinct phase for this idea, and have added it in as a general descriptor in the abstract (line 12) but have continued to use cross-ratio afterwards since we are comparing explicitly to Lievens et al. (2022).

4. L.13 remove "cross".

   We have clarified that this CCPR is called Cross-Ratio in line 12 and use that term here to match the current literature usage.

5. L.14-15: I would reword the last part of the sentence. It is not clear to me is there is or not a correlation/relationship.

We have reworded this sentence for clarity. This sentence now reads: "We find the cross-ratio increases through the time series for snow depth over ~1.5~m but that the cross-ratio decreases for snow depths less than ~1.5~m."

6. L.16: correct "frame work" to "framework".

   Corrected

7. L.24: I would change "the defining hydrologic variable of the seasonal snowpack" to something like: "an important hydrological variable of the seasonal snowpack". It's not the only important hydrological variable for water management.

   There are definitely other important variables for water management. Changed to "one of the defining…"

8. L.33-34: I would break this sentence into two parts. I would put the correlation length part in the second sentence. In terms of correlation length, I imagine it relates to snow surface spatial auto- correlation. Correlation lengths is used in many ways in snow remote sensing, a bit more detail would be useful here.

   We have split this sentence into three sentences to improve readability. We have also changed "correlation length" to "spatial autocorrelation" to clarify our meaning. This now reads: "Networks of in-situ weather stations (e.g., SNOTEL in the United States) make point measurements of snow depth with high temporal resolution. However, accurate spatial interpolation required to generate distributed products presents a significant challenge (Dressler et al., 2006; Bales et al., 2006; Schneider and Molotch, 2016). This challenge is largely due to snow's typical spatial autocorrelation length of 50–200 m (Trujillo et al., 2009)." (line 31-34)

9. L.36: I would specify here that it's for the NASA SWE product. The algorithm uses 19 and 37GHz where the 37 GHz saturates but the lower frequencies of AMSR-E and AMSR2 are sensitive to deeper snow than this.

   Added "using the typical 37 GHz" (line 37)

10. L.52: I feel like the paragraph with the overview of SAR methods should site the review paper of Tsang et al. (2022) which is cited further in the paper.

    In response to reviewers' comments we have now added a paragraph discussing other SAR-based snow estimation techniques and relevant review papers including Tsang et al. (2022). Please see our response to Reviewer #1, comment 4 and lines 62-71.

11. L.53: change "a type of" to "an".

   Changed.

12. L.57: add "and polarization" to "SAR signal's frequency".

   Added.

13. L.57: replace""to"to retrieve".

   Replaced.

14. L.67: delete "a".

   Deleted.

15. L.72: no need to specify the frequencies here but it can span further than 40 GHz

   We have changed this range from "1-300 GHz" (line 83)

16. L.74: Needs a textbook reference here.

   Added textbook references (Ulaby 1980, Long, 1975), tower comparisons (Cihlar and Ulaby 1974, Naderpour et al. 2022, Branger et al. 2023) (line 87, 88, 92, 93, 96)

17. L.75: The citations are in a sentence alone.

   Fixed.

18. L.79: delete substantial.

   Deleted.

19. L.79: replace "the depth of microwave penetration" to "microwave penetration depth".

   Replaced.

20. L.96: there is also second order scattering, i.e. the impacts of multiple scatterers from multiple interfaces in different orientations (roughness) which increases the cross-polarization backscatter. This effect could be highlighted by a thicker snowpack.

Added this to the list of potential effects leading to higher depolarization. (line 123)

21. L.97: these anisotropic clusters are rarely at the surface and are created within the snowpack via metamorphism. It requires a strong temperature gradient which can be amplified by a thicker snowpack. A good reference for the impacts of the anisotropic snow grains is the publication of Picard et al. (2022) on "microwave snow grain size".

We have changed "new snow" to "snow depth increases" and included this citation. (line 126)

22. Figure 1: Not exactly how a SAR system works, here it seems like the emitter and receiver are on two different platforms. Also, the volume scattering at C-Band is a very small portion of the backscattered signal. The other components should be highlighted.

We agree this original figure was unclear. We have separated the two time steps to clarify that we are discussing a monostatic configuration and highlighted some other components of volume scattering (vegetation and ground) to this conceptual figure. Also changed the caption to clarify this figure is showing an idealize conceptual figure.

23. L.103: Again, the cross- to co-pol ratio is more intuitive to me.

Sticking with cross-ratio since that is the term used in the literature.

24. L.122: IW could in theory be HH+HV but VV+VH is the preferred polarizations over land. I would rephrase: [...] IW swath mode, dual-polarized vertical transmit, and vertical/horizontal receive (VV+VH).

Rephased.

25. L.130: multi-looking is used in many different flavors for SAR applications. Do you mean a 3x3 block average, which reduces the 30m resolution to 90 m resolution?

Yes, we feel that unless otherwise specified multi-looking refers to this block averaging.

26. Figure 2: The boxes are small. Subplots of close ups of the three bigger AOIs would be more useful here, i.e. three maps and their bounding boxes could be shown in the broader map at the top right.

We have experimented with these subplots but want to maintain the full view of the study sites presented.

27. L.163-164: These lines seem out of place. Delete?

Agreed, we have deleted these lines.

28. L.180-181: The coefficient values are results and could be discussed with regards to the Lievens parameters. I would move this to the results section in a new section on the parameterization. Some info from the appendix could also be included in this new section.

We originally had this configuration but after deciding that we felt that a whole new algorithm was the best approach going forward, we decided to move the algorithm discussion to the appendix to avoid distracting readers with too many details on an algorithm we ultimately suggest is inadequate to capture the snow information in the S1 backscatter. (see response to major comment #3.

29. L.186-195: This section should be in results/discussion Table 2: I would include the nRMSE as well.

While an analysis of the differences between the stated algorithm in Lievens et al. (2022) and the current CSNOW product would be an interesting comparison we feel it would distract from the overall focus of this paper (how the published methods in Lievens et al. (2022) and the Sentinel-1 CR values relate to lidar snow depths) to move these into the results and Table 2.

30. L.233-234: Given the empirical nature of the retrieval method, it would be useful to determine what is the fraction of pixels that belong to the different "classes". This alone could explain these results if you have an imbalance dataset where most pixels/measurements have deeper snow, moderate FC, and are at higher elevation.

Agreed. Some of this analysis is discussed in Appendix B when we look at the impact of FC on parameterizations. We know that the results for deeper snow are not biased by the class distribution since the histograms in Figure 3b show we have generally lower snow depths but a lower nRMSE at higher snow depths.  Added histograms to show distributions to figure 6 to help

clarify that we don't have imbalanced dataset affecting these values. Since elevation quantile and spatial resolution are both even distributions we omitted them for figure space.

31. Not surprised with the coarser resolution since you remove the spatial variability of snow and landscape properties.

    Agreed. See discussion of this very idea (lines 334-341).

32. L.241-243: This needs to be quantified and not just analyzed qualitatively.

    We have quantified for the lidar which has a much larger dataset than these sparse SNOTEL sites (lines 273-274)

33. L.246-248: Also needs to be quantified.

    Added in text quantifying this relationship (line 273-274).

34. Figure 8: This figure shows more than what is discussed here... The fact that <0.5 m you have no change in deltaCR indicates that you have no significant change to the signal coming from the snow. Then the drop seems to indicate that there is attenuation of the deltaCR with .5 to 1m of snow. Finally, with 1m of snow and more, there seems to be an added contribution to the deltaCR (volume scattering?). That figure in itself is not sufficient to explain the different scattering mechanism in action. The same figure but for VH and VV separately would help identify which mechanism dominates and its link to snow depth.

    Added in both VH and VV boxplots along with mean values and quantified relationship for all in text. Also added in a discussion for the new subpanels in results (lines 276-281).

35. L.262: It's the first time SNR is mentioned. It's an important aspect of SAR retrieval algorithm. I imagine you mean an increase in SNR for VH (VV should be fairly stable in dry snow conditions).

    Added in clarifying definition for SNR (increased snow depth change related signal relative to other sources, such as thermal noise or radar speckle). (line 295)

36. L.263-266: No-correlation needs to be quantified.

    Added in correlations for 0-1 meter group and 2+m group. (line 297-298)

37. L.267: I would even reduce it to snow-air and snow-soil interfaces since the dielectric contrast between two snow layers is very low, and vertical polarization is less sensitive to interface reflectivity than horizontal polarization.

Intersnowpack layer interfaces have been shown to cause depolarization and VH backscatter (see Brangers et al. 2023). Edited to clarify we are discussing within the snowpack and added in anisotropic grains and grain clusters. (line 301-303)

38. L.270-271: This should be included in the introduction with the SAR sensitivity to snow.

We think this makes more sense in the discussion of challenges of this specific approach rather than in the SAR theory section which focuses more on general concepts to acquaint the reader rather than tell them specific challenges of this technique.

39. L.277-278: This could be due to the low number of sampling points with low FC to retrieve the new empirical parameters of Eq. 1.

See the added histograms in Figure 6a, b,c to see we have a pretty even distribution of FC that seems to be unrelated to improvements in algorithm performance.

40. L.282: SNR could be a way to mask out some pixels. Wet snow will usually show a very low VH signal which, depending on the surface roughness, would be close to the noise floor.

This is definitely true. There is a wet snow masking algorithm that is implemented based on Lievens et al. (2022) and addressed in appendix B.

41. L.307-308: I was under the assumption that the parameters were fixed using the 90m scale data. This second point seems to indicate that the parameters were changed for different resolutions. Please clarify.

The parameters are fixed based on the 90m scale data. This discussion has to do with the impacts of using a C parameter at 90m resolution to minimize MAE and then as you coarsen you tend to decrease the spread of the distribution leading to a better fit of the data.

42. L.322: Not necessarily true, this is true for passive microwave signals. Madore et al. (2023, see chapter 6) have shown that K-Band signal could be sensitive to several meters of snow which is at a higher frequency than Ku-Band.

We could only find this paper by Madore et al. (2023) ([https://www.researchgate.net/publication/372886204_Temporal_Analysis_of_Snow_Stratigraphy_and_Melt-Freeze_Crusts_Using_a_24_Ghz_Frequency_Modulated_Continuous_Wave_Fmcw_Radar_in_Avalanche_Terrain](https://www.researchgate.net/publication/372886204_Temporal_Analysis_of_Snow_Stratigraphy_and_Melt-Freeze_Crusts_Using_a_24_Ghz_Frequency_Modulated_Continuous_Wave_Fmcw_Radar_in_Avalanche_Terrain)) and it seems to be a preprint at this point with no chapter 6. Could you please provide more information?

43. L.331: Retrieving eq. 1 parameters on dry snow conditions tested? or all data was used.

     All data was used for parameters optimization.

44. L.336: but there seems to be a detectable decrease between <.5 and .5-1 m of snow. Any possible explanation?

     Included 3 possible explanations. (line 374-376)

45. L.343: That is a major weakness of this algorithm. It could be included in the introduction since it was mentioned by most reviewers publicly in the original Lievens paper in The Cryosphere.

     It is included in line 105 that our understanding of C-band - snow interactions is poorly understood. Though it is not specifically called out as a weakness.

46. L 350: Including passive microwaves is very difficult in high topography areas with its coarse resolution.

     Yes, it would certainly only be part of the solution which is the reason we suggest multiple possible solutions and the need for more research.

47. L.350: I would also include polarimetric SAR approaches.

     Included (line 395).

48. L.353-354: I agree that this is more important than trying to tune an empirical algorithm for the infinite number of landscapes we can find globally.

     Thank you for these comments. They have certainly improved the writing and analysis. Especially the suggestion to include delta VV and VH in the boxplots was well worth the work.

---

## Author Comment (AC4)

June 21, 2024
Evaluating Snow Depth Retrievals from Sentinel-1 Volume Scattering over NASA SnowEx Sites
By: Z. Hoppinen et al.

Reviewer comments are shown in black. Responses are in blue.

**Response to Reviewer #1**

**General Comments:**

A novel method to retrieve snow depth from C-band Sentinel-1 observations was previously introduced by Lievens et al. (2019). While potentially very valuable as a novel source of snow information, the study has also caused a degree of debate in the community due to the conventionally assumed insensitivity of radar backscatter at this frequency to snow accumulation. Although new theoretical analysis suggests that such sensitivity may be possible in certain cases indirectly through the sensitivity of cross-polarized backscatter to anisotropic snow structures, an independent validation of the method beyond some case studies has been largely lacking. The study by Hoppinen et al. attemps just this, providing a potentially valuable contribution to the community, also by giving access to an open-source software replicating the snow depth retrieval method.

The study itself is well written and clear. I find the presentation of results convincing and thorough. I recommend publication of the paper, after considering the following few minor suggestions.

We thank the reviewer for these insightful comments. Accordingly, we have modified the manuscript with the changes highlighted below and believe that the manuscript is much improved with these corrections.

**Minor comments**

1.  Abstract line 8: 2020-2021

    Changed from 2020-21 to 2020-2021

2.  Abstract line 9 and throughout the paper; "poor agreement". While I agree with the authors that the agreement is indeed poor, I would still suggest another way of pointing this out. Also, I'm not sure anyone has quantified what is "poor"… e.g. just stating the R value and nRMSE should be sufficient for readers to make their own conclusions. You can always cite requirements placed on e.g. nRMSE, as you have done in several places.

Also you could cite values obtained with PMW (e.g. Mortimer et al., 2022). In the abstract you could just say the achieved accuracy "is considerably lower than requirements placed for remotely sensed observations of SD" or something similar.

We appreciate this suggestion and agree that "poor" is somewhat vague. We have modified this section of the abstract to read "Across all sites, we find agreement between the Sentinel-1 snow depth retrievals and the lidar snow depth measurements to be considerably lower than requirements placed for remotely sensed observation of snow depth, with a mean RMSE of 0.92 m and a mean Pearson correlation coefficient R of 0.46."

3. Introduction, line 36 "passive microwave measurements saturate" only applies to the 37 GHz frequency typically used in retrievals. Please reword.

Added "at the typically used 37 GHz" (line 37)

4. Introduction, around line 60. Here, it would be appropriate to briefly acknowledge the diverse methods SAR could be used to retrieve SD/SWE: 1) repeat-pass InSAR at low frequencies to obtain deltaSWE 2) single-pass InSAR (DEM differencing) to obtain SD 3) volume scattering approach to obtain SWE directly. Please add also a few appropriate references. You can then tie Lievens et al. more or less to the volume scattering approach, giving a motivation for section 1.1. Please also recap some of the difficulties associated with the volume scattering approach, namely the separation of ground and snow backscattering contributions, as well as the influence of snow microstructure.

To address this reviewers concerns, we have added the following new paragraph to the introduction of the manuscript (line 62-71):

"Numerous techniques have been explored to extract snow depth or SWE from SAR imagery. Such techniques include evaluating backscatter changes to retrieve snow characteristics (Ulaby and Stiles, 1980; Bernier et al., 1999; Shi and Dozier, 2000; Chang et al., 2014; Lievens et al., 2019), using change in travel time information between image acquisitions to approximate SWE changes (Guneriussen et al., 2001; Deeb et al., 2011; Li et al., 2017b; Dagurov et al., 2020; Marshall et al., 2021; Ruiz et al., 2022; Tarricone et al., 2023; Palomaki and Sproles, 2023; Oveisgharan et al., 2023; Hoppinen et al., 2024), exploiting SAR travel time change sensitivity to local slope to capture SWE (Eppler et al., 2022), differencing DEMs for snow depth (Leinss et al., 2018), using differences in the polarimetric response of radar travel times (Leinss et al., 2014, 2016; Voglimacci-Stephanopoli et al., 2021), using travel time changes of frequency subswaths for SWE estimates (Engen et al., 2004), and utilizing phase noise in SAR imagery for snow coverage (Shi et al., 1997; Singh et al., 2008). More detailed discussions of these techniques are available in Tsai et al. (2019), Awasthi and Varade (2021), and Tsang et al. (2022)."

5. Section 1.1, line 75. Surface scattering contributions for dry snow should be very small compared to volume scatter and ground backscatter. This could be good to point out, perhaps with a reference?

This is true, however surface scattering is also wavelength and incidence angle dependent. Multiple studies of avalanche debris have suggested that surface-roughness factors likely contribute a substantial amount of backscattered energy at C-band. We found this point interesting and have decided instead to highlight the implicit assumption that this surface scattering is relatively unchanged throughout the winter or is correlated with snow depth and hence folded into the algorithm (line 99; Figure 1).

6. section 1.1 line 77 "Some SAR-based methods…" brings up the question of what other methods there are. See previous comment #4, please reword.

Reworded this section to "Higher frequency SAR systems". See the major revisions we have made for this in response to your comment #4.

7. section 1.2, lines 111 and 115, maybe elsewhere: suggest to change "in Lievens et al." to "by Lievens et al."

Changed from "in Lievens" to "by Lievens" as suggested and two other locations.

8. Section 2.2 lines 163-164. Sentence seems out of place/complementary of what comes on lines 165-166. Remove?

Removed.

9. section 2.2 line 174 & elsewhere. I guess the parameters A, B and C are not actually dimensionless? delta SD should come out as meters?

A and B are dimensionless and C has units of m/dB. We have added the following sentence in Section 2.2 to clarify the parameter units: "Note that A and B parameters are dimensionless while C has units of m $dB^{-1}$." (line 193-194)

10. Section 2.2 line 187. I find the rather limited correlation surprising, it is a pity that apparently this could not be pursued further. I'm also a bit lost why the intercomparison between the products produces a R value of 0.64, when comparisons to lidar data are apparently very similar between the two… can you elaborate still on the comparisons to lidar data? Did this represent e.g. a subset of the product intercomparison? Maybe even

some map comparisons between C-SNOW and your retrievals could be in order e.g. in the Appendix?

There exist multiple challenges with comparing our retrievals to the C-SNOW product.

The main challenge with comparing the two snow depth datasets is that the published methods in Lievens et al. (2022), which we follow here as closely as possible from their description in the paper, differ from the actual methods used to generate the data and figures in the paper as well as the data available in the C-SNOW repository (https://ees.kuleuven.be/eng/apps/project-c-snow-data/). This discrepancy was confirmed to us by personal communication with Dr. Hans Lievens which we cite in our manuscript (line 216). Additionally there is little to no documentation or metadata available on C-SNOW to clarify which data version relates most closely to the published 2022 methods.

We have compared our results to two versions ("experimental_sentinel1_snow_depth_data" and "West_US_Canada"), resampled our retrieved snow depths and lidar snow depth maps to the resolution of the C-SNOW products and compared all three products together, applying the stated parameters from Lievens et al (2022). We show here the results from the comparison with the "West_US_Canada" version because it had the highest correlation between our retrievals and the C-Snow datasets. We ran this comparison over all 9 lidar acquisitions and computed site-by-site correlations between the two retrievals and the lidar datasets. With so much uncertainty and our overall conclusion that much more complex algorithms are necessary anyways, we think it would be confusing to show too much of this algorithm intercomparison in the main publication or appendixes.

11. Figure 3 panel b: I guess x-axis label could be just "Snow depth" since both lidar and S1 SDs are presented.

Changed to "Snow Depth"

---

## Referee Report (RR1)

In response to general comment: We also do compare our re-optimized algorithm to in situ SNOTEL data (Figure 7), although the in situ network is less dense across our validation sites than what is available in the Alps.

- These in situ stations are the ones in your LIDAR regions. I would recommend using the entire snotel stations in the Sentinel-1 frames you processed. I assume there should be tens of snotel stations in them. You don't need to just use the in situ stations in LIDAR data.

In response to 3: We disagree that this figure is misleading. In panel 5a the distributions are generated by grouping the lidar snow depths themselves. So we impose strict bounds on the lidar (blue) snow depths and compare the corresponding S1 (orange) snow depths, which have a much wider range of values and therefore wider/shorter distribution curves. When the selected lidar and S1 distributions have similar ranges (e.g. 75-100% forest cover in panel 5b or 500m spatial resolution in panel 5f) the relative height of the distributions is closer. The 25th, 50th, and 75th percentiles are notated with dashed lines in the distributions for a quantitative comparison; we have added this information explicitly to the figure caption.

- It is showing the mean and 75% but the histograms are not normalized. Like I said, the maximum of blue in 5a should be much bigger than the maximum of orange if they are normalized.

In response to 5: Revised this sentence to clarify we are talking about the spatial overlap of S1 swaths from different orbit geometries imaging a point every 2-6 days, while the revisit interval for a matching orbit geometry is either 6, 12, or 18 days. (line 148-150).

Also clarified we used all available images. Appendix A contains details on the separation and normalization of images clarifying that we used ascending, descending, S1A, and S1B.

- Again it is not clear when you are talking about 2-6 days, does it include ascending, descending, S1A, and S1B? If so, it needs to state it here.
- Also, I don't think the statement of "most locations" have 2-6 days acquisition is correct. It would be best if you show a heatmap for the study area to show the mean revisit time. If there is no overlap, you will have 6 days including asc des. If you have overlaps it should decrease to 3 days. And the overlap in mid-latitudes are not for most locations.
- I don't see explaining about all acquisitions in appendix A.
- The sentences explaining this needs some grammatical edits, very hard to read.

In response to 6: We are following the methods described in Lievens et al. (2019, 2022). There is a full description of this choice in Lievens et al. (2019) and in Appendix A of our manuscript.

- If you are explaining something in the text, it should be clear. I do not understand what you did here. You mentioned in order to resolve incidence angle difference, you are subtracting the mean so they all have the same mean? If so, mean of what? Spatial or temporal mean. What do you average? If I understand this correctly, shifting means will affect the CR from snow. I also read the appendix A. Over there it just explained how you manage the wet snow by looking signals with the same orbit configuration, not subtracting the mean.

In response to 9: We have changed to "volume scattering" to "depolarization" (line 294). Added definition of SNR term as we are using it here (lines 250).

- The SNR is used in an incorrect way. I am not sure why we use noise here anyway. If we are talking about signal, we can just say backscattered signal from snow increased or decreased. The new edit makes it even harder to follow.

---

## Author Response (AR2)

**Reviewer Responses Round 3 - Evaluating Snow Depth Retrievals from Sentinel-1 Volume Scattering over NASA SnowEx Sites**

September 12, 2024

We thank both reviewers for their thoughtful comments and consideration. Please find our responses detailed below along with the track changes document included.

**Reviewer 3**

*My only concern still remains with Fig. 1. I would suggest adding a red arrow showing "backscatter" from the snow-ground interface in VV. As it stands, it seems like there is only specular reflection from the ground, thus no VV signal. VV is still the dominant source of backscattered power measured by the sensor between the two polarizations (compared to VH).*

We realized our returning energy having such a steep angle was confusing and could be interperted as specular instead of returning energy. We have adjusted the angle to show returning energy directly going to the monostatic configured sensor and showing specular reflection from only the ground interface. Hopefully this is clearer and we appeciate you pointing this out.

*I would also suggest adding a reference to Borah et al. (Preprint) and Zhu et al. (2023), which supports some of the points in the results/discussion sections.*

We have incorporated these great papers/pre-prints into the discussion in the appropriate locations. Thanks for sharing!

**Reviewer 4**

*These in situ stations are the ones in your LIDAR regions. I would recommend using the entire snotel stations in the Sentinel-1 frames you processed. I assume there should be tens of snotel stations in them. You don't need to just use the in situ stations in LIDAR data.*

This is a great idea for a future analysis but outside the scope of this analysis since we are primarily focused on a comparison against our high quality SnowEx lidar snow depth datasets. The CR analysis against the snotel stations is primarily to inform the reader's understanding of the relationship between the Sentinel-1 cross ratio values and snow depth through time. We feel that future analysis exploring the tens to hundreds of snotels in the Sentinel-1 scenes would be an excellent contribution and have included this suggestion in our "Future work" section (line 412-413).

*It is showing the mean and 75% but the histograms are not normalized. Like I said, the maximum of blue in 5a should be much bigger than the maximum of orange if they are normalized.*

We do not disagree with this statement; however, we are intentionally choosing to compare the raw data throughout our analysis, not the normalized values.

*Again it is not clear when you are talking about 2-6 days, does it include ascending, descending, S1A, and S1B? If so, it needs to state it here.*

We have updated line 159 from "we downloaded all available S1 images..." to "we downloaded all available (ascending and descending, S1A and S1B) S1 images..."

*Also, I don't think the statement of "most locations" have 2-6 days acquisition is correct. It would be best if you show a heatmap for the study area to show the mean revisit time. If there is no overlap, you will have 6 days including asc des. If you have overlaps it should decrease to 3 days. And the overlap in mid-latitudes are not for most locations.*

We have changed this sentence to 2-12 days to address your concerns. We have also included an additional Appendix C table clarifying all acquisition timings, platforms, orbits, and flight directions we used for each site.

*I don't see explaining about all acquisitions in appendix A*

This information is actually in the main text starting on line 160. We have reworded this section to clarify the technique used in Lievens et al. (2022).

*"Also clarified we used all available images" - The sentences explaining this needs some grammatical edits, very hard to read.*

We apologize for the confusion. We were attempting to state that we had added language clarifying that we used all available Sentinel-1 images that fell within the bounding boxes of our lidar acquisitions including ascending, descending, S1A and S1B. Hopefully the revisions made in response to your comments above help to clarify the text in this section.

*If you are explaining something in the text, it should be clear. I do not understand what you did here. You mentioned in order to resolve incidence angle difference, you are subtracting the mean so they all have the same mean? If so, mean of what? Spatial or temporal mean. What do you average? If I understand this correctly, shifting means will affect the CR from snow. I also read the appendix A. Over there it just explained how you manage the wet snow by looking signals with the same orbit configuration, not subtracting the mean.*

We have rewritten these sections (lines 160 onwards) to clarify how we are handing the varying incidence angles between different orbit geometries. Since we also directly reference the same methods described in Lievens et al. (2022) we hope the combination of the clarified writing, that reference, and our open source code will allow readers to understand our methodology.

*"Added definition of SNR term as we are using it here (lines 250)." - The SNR is used in an incorrect way. I am not sure why we use noise here anyway. If we are talking about signal, we can just say backscattered signal from snow increased or decreased. The new edit makes it even harder to follow.*

We disagree that this is an incorrect usage of SNR and believe that our new wording clearly spells out that noise refers to non-snow (ie non-signal of interest) related backscatter signals. We discuss the relative changes in our signal of interest to our non-signal of interest (noise).